# WEIGHT DECAY MAY MATTER MORE THAN $\mu$P FOR LEARNING RATE TRANSFER IN PRACTICE

**Atli Kosson**[1,2,†]   **Jeremy Welborn**[1],   **Yang Liu**[1],   **Martin Jaggi**[2],   **Xi Chen**[1]

[1]Amazon FAR (Frontier AI & Robotics),  [2]EPFL

## ABSTRACT

Transferring the optimal learning rate from small to large neural networks can enable efficient training at scales where hyperparameter tuning is otherwise prohibitively expensive. To this end, the Maximal Update Parameterization ($\mu$P) proposes a learning rate scaling designed to keep the update dynamics of internal representations stable across different model widths. However, the scaling rules of $\mu$P rely on strong assumptions, particularly about the geometric alignment of a layer's inputs with both its weights and gradient updates. In this large-scale empirical investigation, we show that these assumptions hold only briefly at the start of training in the practical setups where learning rate transfer is most valuable, such as LLM training. For the remainder of training it is weight decay rather than $\mu$P that correctly stabilizes the update dynamics of internal representations across widths, facilitating learning rate transfer. This suggests $\mu$P's scaling primarily acts as a form of implicit learning rate warmup, allowing us to largely replace it with modified warmup schedules. Together these findings fundamentally challenge prevailing beliefs about learning rate transfer and can explain empirical observations such as why $\mu$P requires the independent weight decay variant for good transfer.

## 1 INTRODUCTION

The Maximal Update Parameterization ($\mu$P) has become a cornerstone of efficient training at scale, particularly for large language models (LLMs). Many open-recipe LLMs have used variants of $\mu$P (Dey et al., 2023; Hu et al., 2024; Liu et al., 2023; Falcon-LLM Team, 2025; Cohere et al., 2025), and some commercial models with unknown training details likely use similar learning rate (LR) scaling approaches. For example, Llama4 used an internal technique called MetaP, Grok-2 configuration files suggest $\mu$P use, and the GPT-4 report (Achiam et al., 2023) cites a key $\mu$P paper coauthored by OpenAI staff (Yang et al., 2021). It has also impacted theory, showing how feature learning can occur in infinitely wide networks, contrasting with the neural tangent kernel regime (Jacot et al., 2018).

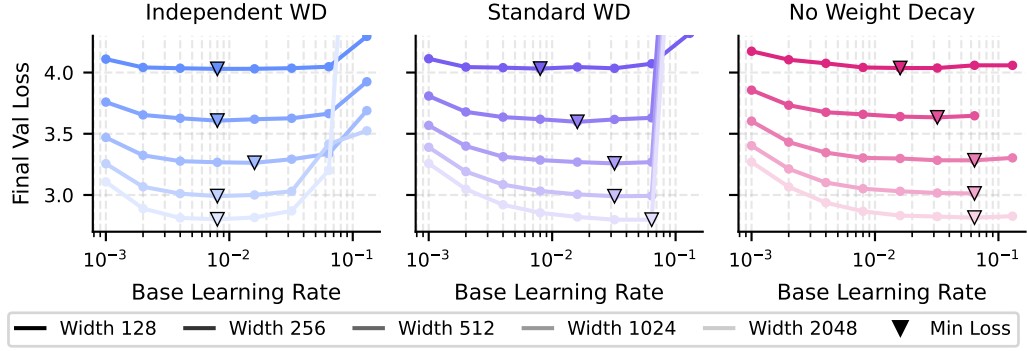

**Figure 1: $\mu$P specifically requires independent weight decay for good learning rate transfer** when pre-training LLaMA networks with AdamW (20B tokens, width 2048 and 1B params, see Appendix F.3). Independent weight decay achieves this by counteracting $\mu$P's update scaling later in training which is needed because the core assumptions $\mu$P's scaling is based on quickly break down. Standard weight decay does not counteract $\mu$P's scaling and consequently results in poor learning rate transfer on longer experiments. No weight decay behaves somewhere in-between, see Appx. B.

---

[†]Work done while interning at Amazon FAR

Despite this success, several works (Wortsman et al., 2024; Blake et al., 2025; Wang & Aitchison, 2025; Bergsma et al., 2025) have empirically found that learning rates only transfer well in practice when $\mu$P is combined with the independent variant of weight decay (WD), see Figure 1. In the conventional AdamW formulation (Algorithm 1), weight decay multiplies the weights by $1 - \eta\lambda$ at each step where $\eta$ is the learning rate and $\lambda$ the weight decay coefficient. With independent weight decay, the weights are decayed by $1 - \lambda$ which is not affected by $\mu$P's learning rate scaling. Equivalently, $\lambda$ can be scaled to keep the product $\eta\lambda$ constant as $\eta$ is scaled in the formulation above.

In this work, we demonstrate that independent weight decay is not merely a helpful addition but a central mechanism for enabling learning rate transfer in practice. We begin by developing a unifying framework for weight decay and $\mu$P based on relative updates, the size of updates as a fraction of the current weights or features (Section 2). Within this framework, we show that independent weight decay directly counteracts and ultimately overrides $\mu$P's scaling during training (Section 3). We then show why this is crucial: for the majority of training, it is independent weight decay, not $\mu$P, that correctly stabilizes feature learning across different model widths (Section 4). This stabilization of the feature learning seems to be what facilitates learning rate transfer and is a key objective of $\mu$P.

The need to override $\mu$P's prescribed update scaling arises because the underlying alignment assumptions fail to hold in practice. We show this can happen when the batch size is large relative to the model width, something that commonly happens in practice unlike the infinite-width limit that inspired $\mu$P (Section 5). This analysis fundamentally reframes $\mu$P's practical role as an implicit learning rate warmup. We empirically measure this effect (Section 6) and demonstrate that the practical benefit of $\mu$P can largely be replicated by using stronger, explicit warmup schedules (Section 7). Together, our findings challenge prevailing beliefs about learning rate transfer and provide guidance for achieving robust transfer in practice. In summary our core contributions are:

- We introduce a **relative update framework** that unifies $\mu$P and weight decay throughout training.
- We demonstrate empirically and analytically that $\mu$P's core **alignment assumptions fail** early on.
- We show **independent weight decay overrides $\mu$P**'s update scaling, stabilizing feature learning.
- We reframe $\mu$P's practical benefit as an **implicit learning rate warmup** and verify this empirically.

## 2 A UNIFYING FRAMEWORK FOR $\mu$P AND WEIGHT DECAY

### 2.1 SINGLE LAYER MODEL AND REPRESENTATION CHANGES

The core $\mu$P and weight decay concepts discussed in this work can be understood from the linear transformation of a fully connected layer in isolation. For an input feature dimension $C$, output feature dimension $K$ and batch size $B$, we define this as:

$$\boldsymbol{Y} = \boldsymbol{W}\boldsymbol{X}, \qquad \boldsymbol{X} \in \mathbb{R}^{C \times B} \text{ (inputs)}, \quad \boldsymbol{W} \in \mathbb{R}^{K \times C} \text{ (weights)}, \quad \boldsymbol{Y} \in \mathbb{R}^{K \times B} \text{ (outputs)} \quad (1)$$

A weight update $\boldsymbol{W} \mapsto \boldsymbol{W} + \Delta\boldsymbol{W}$ causes the layer's outputs to change as $\boldsymbol{Y} \mapsto \boldsymbol{Y} + \Delta\boldsymbol{Y}$ for the same input $\boldsymbol{X}$. Note that in the context of neural networks, $\Delta\boldsymbol{Y}$ refers to the *local* representation change from this layer alone. The *total* change would include potential changes $\Delta\boldsymbol{X}$ in the inputs arising from the updates of earlier layers. Local changes can be related to total changes as shown by Yang et al. (2023), but we will focus on local ones in this work for simplicity.

### 2.2 ALIGNMENT RELATES WEIGHT MAGNITUDES TO REPRESENTATION SIZES

Central to this work and $\mu$P in general is connecting the size of the *representation changes* $\|\Delta\boldsymbol{Y}\|$ to the size of weight updates $\|\Delta\boldsymbol{W}\|$. We measure this in terms of the Frobenius norm $\|\cdot\| := \|\cdot\|_F$. It is easy for an optimizer to give a specific weight update size $\|\Delta\boldsymbol{W}\|$; Adam (Kingma & Ba, 2015) does this approximately and sign-based optimizers like Lion (Chen et al., 2023) can do this perfectly. However it is ultimately the size of the representation changes $\|\Delta\boldsymbol{Y}\|$ that determines the impact of an update. For example, a weight update that does not change $\boldsymbol{Y}$ at all has no immediate impact. However, controlling $\|\Delta\boldsymbol{Y}\|$ is much harder than $\|\Delta\boldsymbol{W}\|$ and is the core objective of $\mu$P's LR scaling.

We describe the relationship between $\|\Delta\boldsymbol{Y}\|$ and $\|\Delta\boldsymbol{W}\|$ via the *update alignment*, defined as:

$$\alpha_{\Delta\boldsymbol{W}} := \frac{\|\Delta\boldsymbol{Y}\|}{\|\Delta\boldsymbol{W}\|\|\boldsymbol{X}\|} = \frac{\|\Delta\boldsymbol{W}\boldsymbol{X}\|}{\|\Delta\boldsymbol{W}\|\|\boldsymbol{X}\|} = \sqrt{\sum_{k,b} \frac{\|\Delta\boldsymbol{w}_k\|^2}{\|\Delta\boldsymbol{W}\|^2} \frac{\|\boldsymbol{x}_b\|^2}{\|\boldsymbol{X}\|^2} \cos^2 \angle(\Delta\boldsymbol{w}_k, \boldsymbol{x}_b)} \in [0, 1] \quad (2)$$

This is a weighted root-mean-square average of the cosine similarities between input samples $\boldsymbol{X} = [\boldsymbol{x}_1, \ldots, \boldsymbol{x}_B]$ and rows in the weight update $\Delta\boldsymbol{W} = [\Delta\boldsymbol{w}_1, \ldots, \Delta\boldsymbol{w}_K]^\top$. Note that a high update alignment $\alpha_{\Delta\boldsymbol{W}}$ means that $\Delta\boldsymbol{w}_k$ and $\boldsymbol{x}_b$ tend to point in similar directions and will result in larger representation changes $\|\Delta\boldsymbol{Y}\|$ for a given update size $\|\Delta\boldsymbol{W}\|$. We similarly define the *weight alignment* as $\alpha_{\boldsymbol{W}} := \frac{\|\boldsymbol{Y}\|}{\|\boldsymbol{W}\|\|\boldsymbol{X}\|}$ which relates the weight magnitude $\|\boldsymbol{W}\|$ to representation size $\|\boldsymbol{Y}\|$.

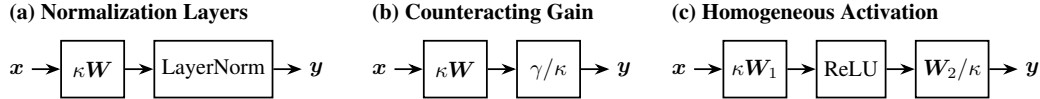

**Figure 2: Neural networks are commonly invariant to certain types of rescaling transformations.**
Examples where applying an in-place scaling factor $\kappa > 0$ does not affect the current block output $\boldsymbol{y}$. Across such transformations, an update with a fixed absolute size $\|\Delta \boldsymbol{W}\|$ has a varying impact on the output while a given relative change $\|\Delta \boldsymbol{W}\|/\|\boldsymbol{W}\|$ always has the same impact. This makes the relative change a more informative measure of the size of updates that better captures their impact.

### 2.3 WEIGHT DECAY MODULATES THE RELATIVE SIZE OF WEIGHT UPDATES

Weight decay is widely thought to mainly function as regularizer despite a long line of work arguing this is not the case in deep learning (Van Laarhoven, 2017; Zhang et al., 2019; Chiley et al., 2019; Li & Arora, 2020; Li et al., 2020; Wan et al., 2021; Li et al., 2022; Kosson et al., 2024b; D'Angelo et al., 2024). Instead it primarily acts as a secondary learning rate hyperparameter that modulates the size of relative weight updates $\|\Delta \boldsymbol{W}\|/\|\boldsymbol{W}\|$. Kosson et al. (2024b) describe these effects in AdamW in detail. In the standard variant (Algorithm 1), weight decay will cause the weight norms to converge to a fixed *equilibrium* value given by $\|\boldsymbol{W}\| \approx \sqrt{KC \cdot \eta/\lambda}$ for learning rate $\eta$, weight decay $\lambda$ and matrix size $K \times C$. At this value, the shrinking effect of weight decay and growth effect of gradient updates balance out, keeping the norm constant in expectation. AdamW's normalization of the weight updates makes $\|\Delta \boldsymbol{W}\| \propto \eta\sqrt{KC}$, which in turn gives relative updates $\|\Delta \boldsymbol{W}\|/\|\boldsymbol{W}\| \propto \sqrt{\eta\lambda}$. Crucially, note that the learning rate $\eta$ and weight decay $\lambda$ affect the relative updates in equilibrium the same way, only their product $\eta\lambda$ matters. In summary:

$$\|\boldsymbol{W}\| \propto \sqrt{KC\eta/\lambda}, \qquad \frac{\|\Delta \boldsymbol{W}\|}{\|\boldsymbol{W}\|} \propto \sqrt{\eta\lambda} \qquad \text{(equilibrium effect of weight decay)} \quad (3)$$

Due to the structure of neural networks, relative changes are often more meaningful than absolute changes (Bernstein et al., 2020; Kosson et al., 2024b;a). A given relative update size $\|\Delta \boldsymbol{W}\|/\|\boldsymbol{W}\|$ has the same impact across a number of equivalent parameter states obtained via transformations like those shown in Figure 2, unlike a fixed absolute update size $\|\Delta \boldsymbol{W}\|$. A similar argument extends to the representations, a given change $\Delta \boldsymbol{Y}$ typically has a greater impact when $\|\boldsymbol{Y}\|$ is small than when it is large. Rescaling transformations do not occur in standard training but weight decay can have a similar overall effect by changing the weight norms over time while normalization layers or learnable gains can undo the effects of this on the output. We show an example of this happening in practice in Appendix G. In this work we focus on relative changes rather than absolute ones based on their informativeness and connection to weight decay, deviating from existing work on $\mu$P. Note that at the start of training these approaches are equivalent since $\|\boldsymbol{W}\|$ and $\|\boldsymbol{Y}\|$ are roughly determined by the initialization. However, later in training they can differ significantly.

### 2.4 FORMULATING $\boldsymbol{\mu}$P IN TERMS OF RELATIVE UPDATES

The primary design goal of $\mu$P is to ensure stable and non-trivial feature learning at any width. Yang et al. (2023) formulate this as $\|\boldsymbol{Y}\|_{\text{RMS}} := \|\boldsymbol{Y}\|_F/\sqrt{KB} = \Theta(1)$ and $\|\Delta \boldsymbol{Y}\|_{\text{RMS}} = \Theta(1)$, i.e., the representations and their changes neither explode nor vanish as the input dimension $C$ grows.[1] In our relative formulation we want to keep $\|\Delta \boldsymbol{Y}\|/\|\boldsymbol{Y}\|$ roughly constant across widths for a given learning rate. The hope is that a fixed relationship between learning rates and representation changes will result in a transfer of the optimal learning rate, but this is ultimately a heuristic without guarantees. We can express the relative representation change in terms of the relative weight update as follows:[2]

$$\frac{\|\Delta \boldsymbol{Y}\|}{\|\boldsymbol{Y}\|} = \frac{\alpha_{\Delta \boldsymbol{W}}}{\alpha_{\boldsymbol{W}}} \frac{\|\Delta \boldsymbol{W}\|}{\|\boldsymbol{W}\|} \tag{4}$$

where we call $\alpha_{\Delta \boldsymbol{W}}/\alpha_{\boldsymbol{W}}$ the *alignment ratio*. To stabilize the relative representation changes $\|\Delta \boldsymbol{Y}\|/\|\boldsymbol{Y}\|$ across different widths $C$, $\mu$P must perfectly counter changes in the alignment ratio by scaling the weight updates appropriately (via the learning rate). This requires knowing how both the update alignment $\alpha_{\Delta \boldsymbol{W}}$ and weight alignment $\alpha_{\boldsymbol{W}}$ behave as a function of the network width $C$.

---

[1]Technically it is the total representation change across training that matters, but Yang et al. (2023) show this can follow from bounded local changes with assumptions about the spectral norms of weights and updates.

[2]We assume matrices are non-zero which is typically the case throughout training, barring special initialization.

At the start of training the weights are randomly initialized. If we assume the weight vector $\boldsymbol{w} \in \mathbb{R}^C$ has independent and identically distributed (IID) zero-mean elements that are also independent of the inputs $\boldsymbol{x} \in \mathbb{R}^C$, we can show that $\mathbb{E}[\langle \boldsymbol{w}, \boldsymbol{x} \rangle^2] = \mathbb{E}[\|\boldsymbol{w}\|^2]\mathbb{E}[\|\boldsymbol{x}\|^2]/C$ resulting in $\alpha_{\boldsymbol{W}} \approx 1/\sqrt{C}$. This is analogous to the computation used to derive variance preserving initialization schemes which call for $\|\boldsymbol{W}\| \approx \sqrt{K}$, which notably does not depend on the input dimension $C$.

The update alignment is slightly more complicated since the update $\Delta \boldsymbol{W}$ is a function of the inputs $\boldsymbol{X}$ rather than independent, causing correlations that tend to increase the update alignment. Specifically the weight gradient is an outer product of the inputs $\boldsymbol{X}$ and the gradients with respect to $\boldsymbol{Y}$. For SGD at batch size one, this makes the update a scaled version of the input, giving perfect alignment $\alpha_{\Delta \boldsymbol{W}} = 1$ regardless of $C$. Based on this we get the core **$\boldsymbol{\mu}$P alignment assumptions**:

$$\alpha_{\Delta \boldsymbol{W}} = \Theta(1), \qquad \alpha_{\boldsymbol{W}} = \Theta(1/\sqrt{C}), \qquad \frac{\alpha_{\Delta \boldsymbol{W}}}{\alpha_{\boldsymbol{W}}} = \Theta(\sqrt{C}) \qquad (5)$$

With this we can scale the learning rate to counteract changes in the alignment ratio in Equation 4. When the network width is multiplied $C \mapsto mC$, we thus get the following **$\boldsymbol{\mu}$P-prescribed scaling**:

$$\frac{\|\Delta \boldsymbol{W}\|}{\|\boldsymbol{W}\|} \propto \left(\frac{\alpha_{\Delta \boldsymbol{W}}}{\alpha_{\boldsymbol{W}}}\right)^{-1} \propto \frac{1}{\sqrt{m}} \text{ (relative update size)} \quad \implies \quad \eta = \eta_{\text{base}}/m \text{ (Adam LR)} \qquad (6)$$

The $\eta$ scaling follows from the relative one based on $\|\Delta \boldsymbol{W}\| \propto \eta \sqrt{KC}$ for Adam and $\|\boldsymbol{W}\| \approx \sqrt{K}$. The optimal *base learning rate* $\eta_{\text{base}}$ is what we aim to transfer across widths, tuning it at a small scale rather than the target scale we are interested in, saving significant compute in the process.

We emphasize that the prescribed relative update scaling in Equation 6 is fully consistent with prior works on $\mu$P and not an artifact of our formulation, see for example Appendix B.3 of Yang et al. (2021), Adafactor "Full Align" in Table 1 of Everett et al. (2024) and Section 5.4 of Yang et al. (2023). Besides scaling the learning rate of hidden layers, $\mu$P involves special handling of the final output layer and attention normalization. Everett et al. (2024) shows these are not necessary for learning rate transfer, combining standard parameterization with layer-wise scaling of the learning rate works equally well or better in practice. We adopt their simpler version in this work although we refer to the scaling as $\mu$P-prescribed or simply $\mu$P for brevity. See Appendix F.2 for details.

## 3 INDEPENDENT WEIGHT DECAY EVENTUALLY NEUTRALIZES $\boldsymbol{\mu}$P'S SCALING

There are two common formulations of AdamW. In the original work by Loshchilov & Hutter (2019), the rate of weight decay was determined by $\lambda$ alone, i.e., at each step the weights are scaled by $1 - \lambda$. However in the default implementation in PyTorch the weights are instead shrunk by $1 - \eta\lambda$. These are equivalent since we can scale $\lambda$ to get one from the other, but will behave differently when $\eta$ is modified by $\mu$P. In this work we base all discussion on the PyTorch variant but consider the following two approaches for weight decay as $\mu$P scales the learning rate $\eta \mapsto \eta/m$:

$$(\eta, \lambda) \mapsto (\eta/m, \lambda) \qquad \text{(standard scaling)} \quad (7)$$
$$(\eta, \lambda) \mapsto (\eta/m, m\lambda) \qquad \text{(independent scaling)} \quad (8)$$

Note that independent scaling keeps the product $\eta\lambda$ constant, meaning the relative updates in equilibrium are not scaled at all (see Equation 3). This fundamentally differs from $\mu$P's prescribed scaling in Equation 6, meaning that **independent weight decay scaling contradicts $\boldsymbol{\mu}$P by eventually overriding its update scaling.** On the other hand standard weight decay scaling results in equilibrium behavior that follows the relative update scaling prescribed by $\mu$P. This is surprising because multiple works have empirically found that using independent weight decay gives better learning rate transfer (Wortsman et al., 2024; Blake et al., 2025; Wang & Aitchison, 2025; Bergsma et al., 2025). We replicate this finding for LLaMA (Touvron et al., 2023) training on DCLM (Li et al., 2024) next token prediction in Figure 1. Together these results suggest that **applying the $\boldsymbol{\mu}$P-theoretical scaling of the relative update size throughout training does not work well in practice.** We validate this finding through direct experimentation in Appendix H.

To the best of our knowledge, this key mismatch between theory and what empirically works has not been pointed out before. In the following sections we use our unified relative framework to clarify the exact roles of both $\mu$P's scaling and weight decay in achieving learning rate transfer in practice.

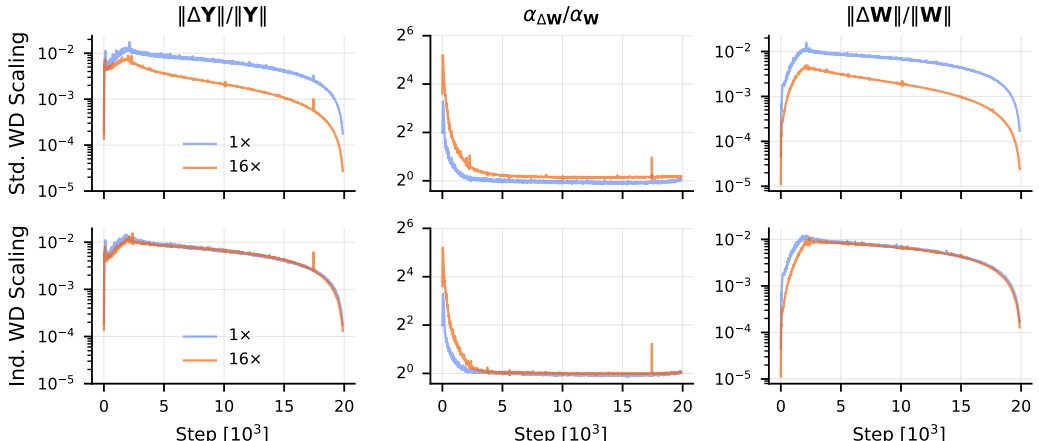

**Figure 3: Standard $\mu$P scaling fails to maintain relative representation changes across widths.** Comparing the relative representation change (RRC) for $\mu$P with standard weight decay scaling (first row) and independent weight decay scaling (second row) for LLaMA training with a 10% linear warmup followed by linear decay. Plots shown for a single representative layer with $C = 128$ at $1\times$ width, see F.3.2 for details. **Left:** Standard scaling does not preserve the RRC for the same learning rate unlike independent scaling. Recall that the RRC is the product of the alignment ratio and the relative weight update (Equation 4). **Middle:** The alignment ratio is similar for both widths and approximately 1. **Right:** To maintain the RRC across widths the relative weight update size should then be kept constant as achieved by independent weight decay rather than $\mu$P's prescribed scaling.

## 4 WEIGHT DECAY COUNTERACTS $\mu$P'S BROKEN ALIGNMENT ASSUMPTIONS

Recall that in our formulation $\mu$P's core objective becomes mapping a given base learning rate to the same rate of relative representation changes across different network widths. In the left column of Figure 3 we experimentally measure how well this is achieved under both standard and independent weight decay scaling during LLaMA pre-training. We find that independent weight decay scaling successfully maintains the relative representation changes across widths as indicated by the similarity of the traces on the lower left panel. In contrast, standard weight decay scaling fails to do this; the same base learning rate results in vastly different relative feature updates for the narrow and wide networks later in training. Since learning rates transfer well with independent weight decay (Figure 1), this suggests that our goal of **maintaining relative representation changes across widths can be an effective objective for learning rate transfer, but $\mu$P relies on independent weight decay to achieve this later in training**.

The reason independent weight decay helps maintain relative representation changes can be understood from the middle and right columns of Figure 3. Recall from Equation 4 that the relative representation change is the product of the alignment ratio shown in the middle column and the relative weight change in the right column. We see that the alignment ratio quickly falls from the width-dependent initial value to approximately one, violating $\mu$P's assumptions. Once this happens, **relative weight changes should be the same across both network widths in order to produce the same relative representation changes, which is exactly what independent weight decay achieves** in practice, matching theoretical models. Standard weight decay scaling results in smaller relative updates for the larger network, as prescribed by $\mu$P. This fails to maintain the relative representation change with the lower alignment ratios that dominate training, preventing learning rate transfer.

In Appendix B we explore the effects of not using weight decay at all. Somewhat counterintuitively, this also results in similarly sized relative weight updates across widths like with independent weight decay. The reason is that scale-invariant weights grow perpetually when updated without weight decay. Once the cumulated updates outweigh the initialization, the weight norm becomes proportional to the (peak) learning rate used, making both the weights and their updates proportional to it. This means their ratios, **the relative weight updates, eventually lose their dependence on the peak learning rate without weight decay**. We believe this is a key reason learning rates somewhat transfer without weight decay despite $\mu$P's broken assumptions. The ever decreasing relative changes, and the lack of control over them, likely also contributes to the loss of performance without weight decay.

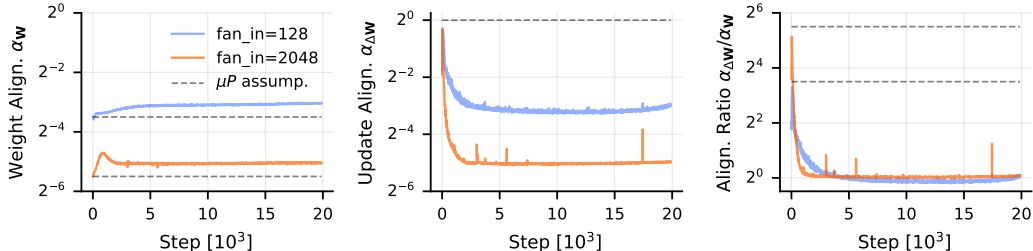

**Figure 4: The alignment assumptions of $\mu$P do not hold in practice**. Measurements of the weight alignment, the update alignment and the alignment ratio for LLaMA training at different widths. The update alignment varies significantly over time, becoming width-dependent. As a result the alignment ratio loses its width dependence. This violates $\mu$P's core scaling assumptions which only hold very early in training. Plots shown for one representative MLP layer, see F.3.3 for details.

## 5 WHY DO $\mu$P'S ALIGNMENT ASSUMPTIONS BREAK DOWN IN PRACTICE?

Figure 4 shows how $\mu$P's alignment assumptions only hold very briefly at the start of training. Both the update and weight alignment can deviate significantly, for the weight alignment this is more prominent in our ResNet experiments (see Figure 13, Appendix E). In this section we investigate why and when $\mu$P's assumptions about each alignment component break down in practice.

### 5.1 UPDATE ALIGNMENT

In Appendix C we show how **the update alignment can become width-dependent when the batch size $B$ is large compared to the input dimension $C$, violating $\mu$P's assumptions.** This is frequently the case in practice unlike in the infinite width setting $\mu$P was originally developed for.

To understand the conceptual reason for the width-dependence, let us look at the simplified case of SGD for a single neuron i.e., $K=1$ in Equation 1. Assume the input samples $\mathbf{x}_b \in \mathbb{R}^C$ and the output gradients $\mathbf{y}'_b \in \mathbb{R}$ are all zero-mean IID random and mutually independent. The gradient for each sample is then given by $\mathbf{g}_b = \mathbf{x}_b \mathbf{y}'_b$ and is also IID and zero-mean. The weight update is given by $\Delta \mathbf{w} = -\eta \frac{1}{B} \sum_b \mathbf{g}_b$, allowing us to write the $i$-th element of the output change $\Delta \mathbf{y}$ as:

$$\Delta \mathbf{y}_i = \langle \Delta \mathbf{w}, \mathbf{x}_i \rangle = \langle -\eta \frac{1}{B} \sum_b \mathbf{x}_b \mathbf{y}'_b, \mathbf{x}_i \rangle = -\eta \frac{1}{B} \mathbf{y}'_i \langle \mathbf{x}_i, \mathbf{x}_i \rangle - \eta \frac{1}{B} \sum_{b \neq i} \mathbf{y}'_b \langle \mathbf{x}_b, \mathbf{x}_i \rangle \quad (9)$$

The output change for each input involves one *self-contribution* term and $B-1$ *interference* terms from other samples. Although random alignment weakens each interference term by a factor of $1/\sqrt{C}$ relative to the fully-aligned self-contribution, the random sum across all $B-1$ terms amplifies their impact by roughly $\sqrt{B}$. As a result, interference dominates when $B \gg C$, giving an overall dependence on $C$. Conversely, for $B \ll C$, the interference becomes negligible making the update alignment invariant to $C$. In models like Transformers and CNNs, the spatial or sequence dimensions act similarly to the batch size. For our LLaMA experiments, the *effective batch size* becomes the total number of tokens per batch (1,048,576), which far exceeds the network width ($C \leq 3 \times 2048$).

In practice the gradients of different samples are not uncorrelated. High initial correlation results in a large update alignment that resembles the small batch-to-width setting that $\mu$P targets. The correlation decreases over time, resulting in width-dependence that violates $\mu$P's assumptions, see Appendix C.

### 5.2 WEIGHT ALIGNMENT

The initial weight alignment matches $\mu$P's assumptions, but analyzing it later in training is hard. Intuitively, we note that we can decompose the weights $\mathbf{W}_t$ at time $t$ into weight updates $\Delta \mathbf{W}_\tau$ from prior time steps $\tau$, where we assume $\Delta \mathbf{W}$ does not include the weight decay term (if used):

$$\mathbf{W}_t = (1 - \eta\lambda)^t \cdot \mathbf{W}_0 + \sum_{\tau=0}^{t-1} (1 - \eta\lambda)^{t-\tau} \cdot \Delta \mathbf{W}_\tau \qquad \text{(update composition of weights)} \quad (10)$$

If the decay rate $\eta\lambda$ is large or $\|\mathbf{W}_0\|$ is small compared to the updates, the $\mathbf{W}_0$ term quickly becomes insignificant. Afterwards, the weight alignment at time $t$ is determined primarily by the alignment of $\mathbf{X}_t$ and recent update terms $\Delta \mathbf{W}_\tau$. **If the current update and recent updates are similarly aligned with the inputs, the overall weight alignment may approximate the update alignment.** This would explain the alignment ratios $\alpha_{\Delta\mathbf{W}}/\alpha_{\mathbf{W}} \approx 1$ that we often observe in practice (see Figure 21).

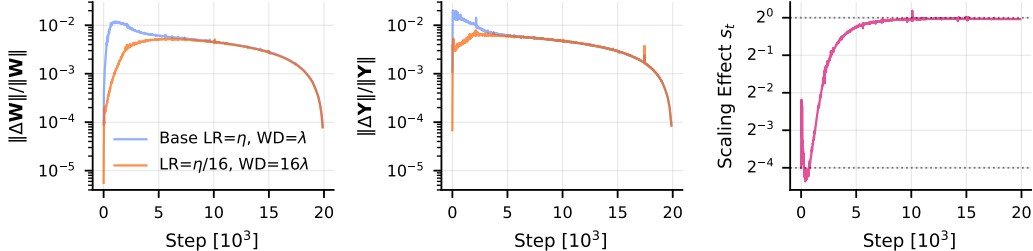

**Figure 5: $\mu$P with independent WD scaling has a warmup-like effect on relative updates.**
Measurements of the relative weight update (left) and relative representation change (middle) for
different learning rate (LR) and weight decay (WD) combinations in a LLaMA layer with 2048
input features. Independent weight decay scaling achieves a given LR-WD product using a lower
LR and higher WD (orange) compared to not performing $\mu$P scaling (blue). This results in smaller
relative updates early in training, similar to an additional learning rate warmup factor. The right
panel shows this scaling effect through the ratio of the relative weight updates for the two runs
$s_t = \|\Delta W_{\text{blue}}\|\|W_{\text{orange}}\|/(\|W_{\text{blue}}\|\|\Delta W_{\text{orange}}\|)$, which goes from $\frac{1}{m} = \frac{1}{16}$ to asymptotically
approaching 1 in a roughly exponential manner. See F.3.4 for experimental details.

## 6    THE WARMUP EFFECT OF $\mu$P WITH INDEPENDENT WEIGHT DECAY

Based on the order-one alignment ratios that we find to dominate practical training, the relative
weight updates should be kept roughly constant across widths. Kosson et al. (2024b) described how
the magnitude of the relative updates in the steady-state only depends on the product of the learning
rate $\eta$ and weight decay $\lambda$ rather than their individual values. However, $(\eta, \lambda)$ pairs with the same
product generally affect the initial phases of training differently. Specifically, high weight decay pairs
like those obtained from independent $\mu$P scaling via $(\eta, \lambda) \mapsto (\eta/m, m\lambda)$ have a warmup-like effect,
where the relative updates at the start of training are comparatively smaller. Therefore, **$\mu$P acts as
a special form of additional learning rate warmup when used with independent WD scaling**.
We measure this effect empirically for LLaMA training in Figure 5, confirming the general effect is
still present despite the lack of the perfect scale-invariance typically assumed by works on weight
decay. Note that since $\mu$P does not scale the updates appropriately for the later phases of training,
**this warmup effect is the only practical benefit of $\mu$P's learning rate scaling for longer runs**.

The shape of the induced warmup effect is generally quite complex. The initial downscaling of the
relative update is always $1/m$ where $m$ is width ratio used for $\mu$P's scaling. The scaling approaches
one over time as the weight norms close in on their equilibrium values. The effective length of the
warmup depends on the initialization value, learning rate and weight decay. In Appendix D we
analyze a simplified setting with a constant learning rate where the scaling ratio becomes:

$$s_t = \sqrt{\frac{1 + (\rho_0^2/\rho_\infty^2 - 1)a^{2t}}{1 + (m^2\rho_0^2/\rho_\infty^2 - 1)a^{2t}}} \underset{\text{if } \rho_0 = \rho_\infty}{=} \frac{1}{\sqrt{1 + (m^2 - 1)a^{2t}}} \qquad \text{(simple warmup effect)} \quad (11)$$

where $a := 1 - \eta\lambda$ is the decay multiplier at each step, $\rho_t$ is the weight RMS, $\rho_0$ is the initialization
RMS value, and $\rho_\infty = \frac{\eta}{2\lambda}$ is the predicted weight RMS in equilibrium. In practice this model does
not accurately predict the shape in Figure 5, overestimating the time required to reach equilibrium.
The reason is likely the interaction of momentum with temporal gradient correlations that do not
appear in the simplified fully random and uncorrelated setting the model is derived for.

Finally we note that $\mu$P's learning rate scaling in AdamW is used to both counteract assumed
changes in the alignment ratio and the size of the relative updates. Recall that the learning rate
is scaled $\eta \propto 1/m$ whereas the relative updates should only scale with $1/\sqrt{m}$. Even without any
assumptions about the alignment ratio, it would make sense to scale the learning rate and weight
decay $(\eta, \lambda) \mapsto (\eta/\sqrt{m}, \sqrt{m} \cdot \lambda)$. This scales $\rho_\infty \propto 1/\sqrt{m}$, matching how variance preserving
initialization scales $\rho_0 \propto 1/\sqrt{m}$. Such scaling preserves both $a$ and $\rho_0/\rho_\infty$ in Equation 11, and
is thus more likely to preserve the relative weight update behavior across training. In comparison
independent $\mu$P scaling has an additional warmup effect of $1/\sqrt{m}$ in the relative updates (which is
desirable), but also decreases $\rho_\infty \propto 1/m$. These smaller weight norms could potentially cause issues
(signal propagation etc.) if there are not learnable gains or normalization layers to counteract them.

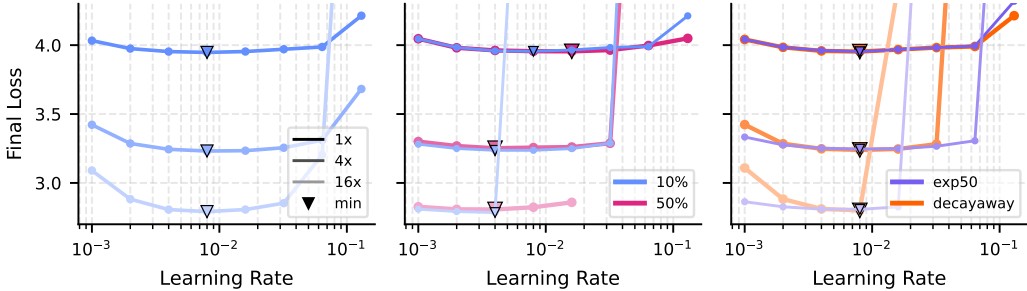

**Figure 6: Stronger warmup can replicate the benefit of $\mu$P with independent WD scaling.** Learning rate transfer plots for LLaMA training using different warmup approaches. **Left:** $\mu$P with independent scaling transfers well. **Middle:** No $\mu$P scaling with either the base 10% linear warmup or a longer 50% warmup. These give a poor learning rate transfer in comparison. **Right:** No $\mu$P scaling with additional warmup incorporating the width scaling factor $m$ can give good transfer, but can require tuning and is less stable at higher learning rates. Additional 50% exponential warmup (Equation 12) compared to a decay-away warmup (Equation 13). See F.3.5 for experimental details.

## 7 CAN STRONGER LEARNING RATE WARMUP REPLACE $\mu$P'S LR SCALING?

Learning rate warmup is already commonly used in practice, making it unclear whether the additional warmup effect from $\mu$P is really needed. The first two panels of Figure 6 explore this for LLaMA training, comparing the quality of learning rate transfer with and without $\mu$P's scaling. Interestingly, we find that **even long linear warmup does not necessarily give the same benefits for learning rate transfer or stability as $\mu$P with independent weight decay**. This suggests that linear warmup schedules may not be very effective for certain setups despite their widespread use in practice. In Appendix E we conduct a similar experiment finding that **the additional warmup effect is not needed for ResNet training**. This could be a part of the reason $\mu$P like learning rate scaling did not appear until transformers became popular, well-normalized convolutional networks may not need it at reasonable scales.

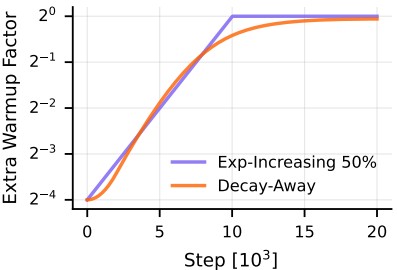

**Figure 7:** Additional warmup factors applied on top of a linear decay schedule with 10% linear warmup for $m = 16$, $\eta = 0.004$, and $\lambda = 0.1$, matching the minimum in the right panel of Figure 6.

We conjecture that linear warmup schedules may increase too fast early on, not allowing enough time for the alignment ratio to fall. Taking inspiration from the more exponential warmup shape from Equation 11, we experiment with two types of additional multiplicative learning rate warmup factors:

$$\hat{s}_t = m^{\min(0,\, t/T_W - 1)} \qquad \text{(exp-increasing warmup)} \quad (12)$$

$$\hat{s}_t = \left(1 + (m^2 - 1)\prod_{\tau=0}^{t-1}(1 - \eta_\tau \lambda)^2\right)^{-1/2} \qquad \text{(decay-away warmup)} \quad (13)$$

Both scale the initial learning rate by $1/m$, similar to $\mu$P, and eventually become one. The exponentially increasing one explicitly defines the length of the warmup through a hyperparameter $T_W$ which can differ from the length of the linear warmup. The decay-away warmup comes directly from the simplified case of Equation 11. In this case the length of the warmup effect differs depending on the value of $\eta\lambda$, which can vary over time if the learning rate changes according to a schedule $\eta_t$.

The final panel of Figure 6 shows the effect of applying these additional warmup factors on top of the existing 10% linear warmup. This works well, offering better stability and learning rate transfer than with the linear schedules suggesting: **Additional exponential warmup can sometimes replace $\mu$P's learning rate scaling in practice.** Although the effect from independent scaling still seems to stabilize higher learning rates slightly better, these results further point to the warmup effect being $\mu$P's primary benefit. We note that in our experiments we applied the warmup factors to all parameters. This differs from the warmup effect of the independent scaling which only affects parameters whose learning rate is scaled by $\mu$P, i.e., the weight matrices of the hidden and final output layers.

## 8   RELATED WORK

We discuss related work in more detail in Appendix A, only briefly mentioning the strongest influences and most closely related works here. Our work adapts the layer-wise LR scaling approach of Everett et al. (2024) and uses a similar alignment definition, although they only consider weight alignment. We also take inspiration from the simpler more accessible approach to $\mu$P introduced by Yang et al. (2023) to isolate the key alignment assumptions of $\mu$P and work with local changes. Finally, we build closely upon the framework for weight decay introduced by Kosson et al. (2024b) and the additional observations about relative updates and learning rate warmup from Kosson et al. (2024a).

The question of why $\mu$P requires independent weight decay was explored before by Wang & Aitchison (2025) who also empirically demonstrated its necessity for learning rate transfer. Their analysis revolves around approximating AdamW as an exponential moving average (EMA) over past updates, which has a form similar to the geometric sum in Equation 10. From this perspective the decay rate defines a time scale for the EMA (analogous to a half-life) which is an alternative characterization of the relative update size in equilibrium. They argue the time scale should intuitively remain constant across network sizes in order for the final network parameters to form a similar average over different data points, but do not justify this formally. We note that although the timescale interpretation can be a useful mental model and approximation, it does not account for the dependency of later updates upon earlier ones. This dependency is exactly what prevents optimization from being performed in a single step by simply taking an appropriate weighted average over all the data. We take a different approach by relating weight decay directly to the goals of $\mu$P through the rate of feature changes.

## 9   DISCUSSION & CONCLUSION

Originally $\mu$P was proposed as a theoretical framework that aims to control feature learning in the infinite width limit at initialization. This work expands upon this by studying feature learning throughout training in practical finite-width networks through a more empirical approach. Our relative formulation provides a unified view of $\mu$P and weight decay, explaining how both play a role in stabilizing feature learning at different times during training. Crucially, this reveals that $\mu$P's downscaling of the relative weight updates for wider networks is only desirable early in training. As training goes on, $\mu$P's scaling must be counteracted by independent weight decay or similar mechanisms in order to stabilize feature learning and thereby facilitate learning rate transfer.

In this work we have focused exclusively on AdamW for two key reasons. First, AdamW is still the most common optimizer for LLM training, a key application area for $\mu$P and learning rate transfer. Second, the original $\mu$P formulation for SGD does not scale the learning rate for hidden layers, using special multipliers instead. This can make standard weight decay behave like independent weight decay under $\mu$P's scaling, making it harder to show the discrepancy. However, we believe the alignment dynamics we have described and $\mu$P's broken alignment assumptions occur in SGD and similar optimizers as well. Matrix-level optimizers on the other hand could significantly change these dynamics. For example, Muon (Jordan et al., 2024) and Scion (Pethick et al., 2025) can result in a constant low update alignment that does not vary over time, allowing them to bypass much of the complexities we have discussed in this work. This could potentially explain their reduced need for learning rate warmup and makes it a promising alternative to $\mu$P for effectively controlling feature learning in practice.

In Appendix E we show that our findings mostly generalize to ResNet training on ImageNet with AdamW. The main difference there seems to be a potentially confounding regularization benefit from higher learning rates. As the network widths increase, smaller relative representation changes minimize the training loss while larger changes minimize the validation loss. Despite this, independent weight decay still gives much better transfer than standard weight decay.

The complex effects of weight decay are unfortunately often overlooked in optimization research for deep learning. The use of weight decay allows specifying the size of relative updates twice, once at the start of training through the learning rate and again in equilibrium through the product of the learning rate and weight decay. Interestingly, $\mu$P with independent weight decay scaling actually makes use of this effect to create the beneficial implicit learning rate warmup. Although we showed a similar effect can be achieved via modified learning rate schedules, $\mu$P with independent weight decay scaling is a perfectly valid and practical way of achieving this. We hope our work can inspire confidence in this approach in practice, provide useful conceptual insights into learning rate transfer and weight decay, and finally highlight a promising way forward via matrix-level optimizers.

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

## LLM STATEMENT

LLMs were used for aiding and polishing writing as well as creating and modifying plotting scripts.

## A EXTENDED RELATED WORK

Adam (Kingma & Ba, 2015) is a widely used optimizer in deep learning. It was originally used with $\ell_2$-regularization but Loshchilov & Hutter (2019) empirically found that decoupled weight decay works better, resulting in AdamW. The version of AdamW we use is shown in Algorithm 1 and corresponds to the most popular variant, the default in PyTorch (Paszke et al., 2019).

Weight decay is unfortunately still widely believed to act primarily as an explicit regularizer despite a long line of work that argues this is not the case in deep learning (Van Laarhoven, 2017; Zhang et al., 2019; Chiley et al., 2019; Li & Arora, 2020; Li et al., 2020; Wan et al., 2021; Li et al., 2022; Kosson et al., 2024b; D'Angelo et al., 2024). Instead weight decay modulates some notion of an "effective" learning rate, which can be seen as measures of the relative update size. Our work builds closely upon Kosson et al. (2024b), who described the equilibrium dynamics of weight decay in detail for AdamW. This included originally noting the scheduling effects, how weight decay can be replaced by controlling the relative weight update, and how the balanced equilibrium behavior of AdamW explains its effectiveness over Adam with $\ell_2$-regularization.

Yang & Hu (2021) proposed the Maximal Update Parameterization $\mu$P as a way to ensure stable feature learning in infinitely wide networks. Yang et al. (2021) demonstrated how $\mu$P allows transferring the learning rates across different network widths and how this is beneficial for hyperparameter tuning. Later works have expanded $\mu$P to network depth rather than only width (Yang et al., 2024; Dey et al., 2025). The original formulation of $\mu$P is quite theoretical but Yang et al. (2023) show a simplified and more accessible derivation based on spectral norms which we base our discussion on. Dey et al. (2024) provide a useful overview of $\mu$P.

Everett et al. (2024) show hyperparameter transfer is possible for other parameterizations than $\mu$P, including standard initialization schemes without the special multipliers used in $\mu$P, using analogous learning rate scaling. We adopt this approach in our experiments as it is simpler and results in better performance according to Everett et al. (2024). They also describe how the learning rate scaling depends on alignment metrics similar to those we define in Equation 4, but use the total weight change from the start of training rather than a single optimization step. Our findings fit particularly well with one case they consider, the no-alignment setting. Their analysis of AdaFactor (Shazeer & Stern, 2018) with weight-proportional updates in this case predicts that learning rates should transfer without any scaling. This is similar to what we observe in practice once weight decay makes the updates proportional to the weights. We expand upon their work with measurements of the update-alignment, describing the role of weight decay, and how the batch size affects alignment.

Atanasov et al. (2022) studied a different form of alignment between the Neural Tangent Kernel (NTK, Jacot et al. (2018)) and the target labels. This alignment is not directly related to the weight and update alignment as we define them as far as we know, but there could be a connection to the update alignment. The NTK measures the similarity between the gradients of different samples, something we hypothesize the update alignment also depends on through the Signal-to-Noise ratio described in Appendix C.

Haas et al. (2025) is very recent work that investigates why higher learning rates than $\mu$P prescribes can be used. They investigate alignment changes as a potential cause but dismiss them in favor of the role of the loss function, showing that typical cross entropy losses help stabilize the final layer compared to $\ell_2$ based losses. Our alignment measure is based on the current update rather than the total update across training, which might explain the different conclusions.

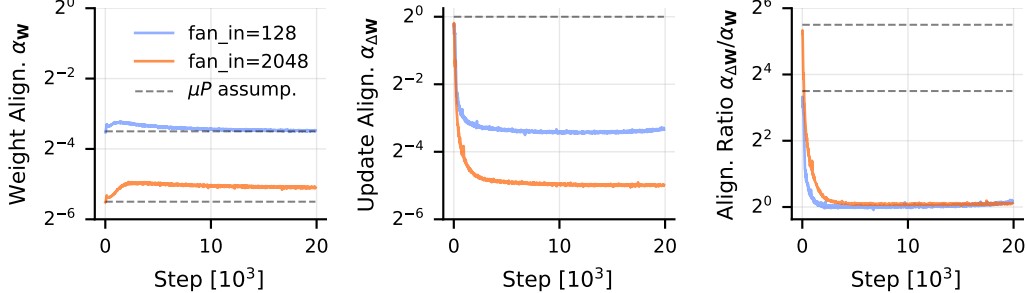

**Figure 8: The alignment assumptions of $\mu$P break down even without weight decay**. Measurements of the weight alignment, the update alignment and the alignment ratio for LLaMA training at different widths. The update alignment varies significantly over time, becoming width-dependent. This violates $\mu$P's assumption which only holds very early in training. A base learning rate of $\eta = 1.6 \cdot 10^{-2}$ is used for the narrower network and scaled down $16\times$ for the wider network via $\mu$P's learning rate scaling. Plots shown for one representative MLP layer, `layers.13.feed_forward.w1`.

# B $\mu$P'S ALIGNMENT ASSUMPTIONS STILL FAIL WITHOUT WEIGHT DECAY

In Figure 1 we observed that no weight decay seems to provide better learning rate transfer than the use of standard weight decay, although it is still not as good as with independent weight decay. The final loss is also slightly worse than the best configuration with either weight decay approach. In this section we explore whether the assumptions of $\mu$P hold without weight decay.

In Figure 8 we see that this is not the case. The alignment ratio still becomes roughly one, independent of the width, meaning that the alignment assumptions of $\mu$P break down exactly like in the weight decay case. **Weight decay is therefore not the cause of $\mu$P's violated assumptions.** This matches our analysis of the update alignment (Section 5) where the width dependence at large batch sizes does not depend on weight decay directly. With an alignment ratio that does not vary across width, the relative updates should be preserved in order to keep the relative representation changes comparable.

In Figure 9 we measure how the relative updates behave across different learning rates. Notably, **without weight decay the peak learning rate has little effect on the size of relative updates later in training**. This could explain why the no weight decay setting seems to be less sensitive to the specific value of the learning rate in Figure 1. Without weight decay the size of the relative updates falls over time compared to when weight decay is used, similar to an extra decay factor in the learning rate schedule. The way that this decay occurs seems to cause the relative weight updates to lose their dependence on the peak learning rate. In Appendix D.2 we show this can happen in a simplified setting for constant learning rates. The conceptual reason is that if the total weight growth dominates the initial value, both the update and the total weight norm have the same proportional dependency on the learning rate later in training. These factors cancel out resulting in roughly identical relative weight updates across peak learning rates. This assumes the network behaves similar to a scale-invariant network, i.e., that there is no strong gradient signal that affects the weight norms. A similar effect of the peak learning rate losing its relevance over time was observed for SGD and normalization layers before (Arora et al., 2019; Mehmeti-Göpel & Wand, 2024).

The optimal learning rate likely becomes a balance between two factors, meaningful learning and stability. Higher learning rates decrease the contribution of the initial weights to the final model, which may allow the model to better learn the distribution. However, high learning rates can also cause large relative updates early in training that can have detrimental effects as explored by Kosson et al. (2024a), perhaps by saturating non-linearities or causing instability. If these detrimental effects occur sufficiently early, the alignment assumptions of $\mu$P may still roughly hold. In this case the optimal learning rate may be decided primarily by early behavior. This may aid optimal learning rate transfer under $\mu$P's scaling, despite its core assumptions only holding briefly at the start of training. We suspect the length of the learning rate warmup may also impact whether the key limiting stability period occurs when $\mu$P's alignment assumptions still hold or not.

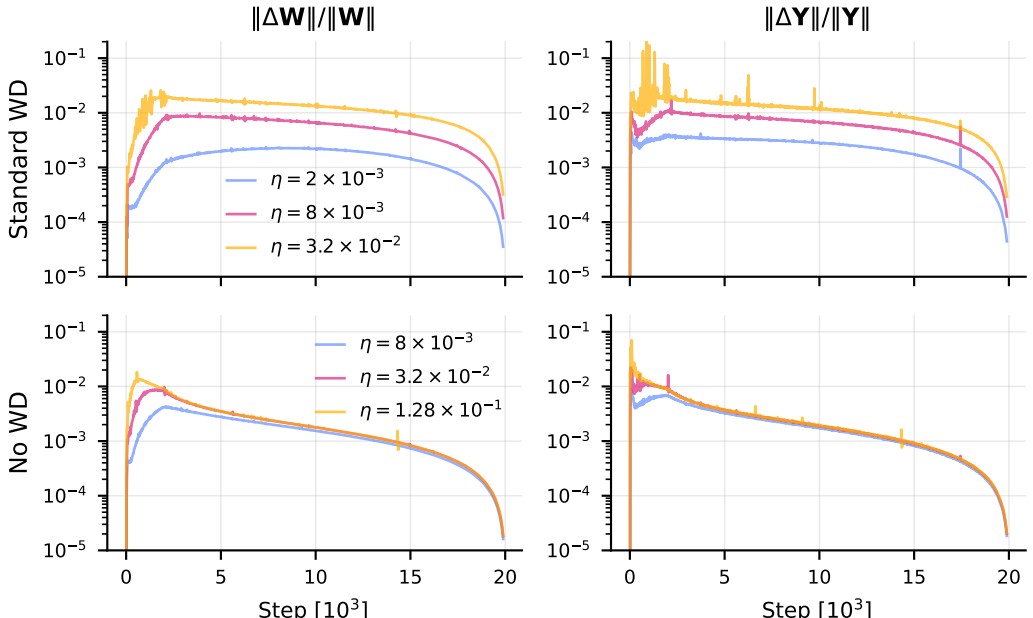

**Figure 9: Without weight decay relative updates become similar across learning rates over time**. Measurements of the relative weight update size and relative representation change for LLaMA training. Plots shown for one representative MLP layer, `layers.13.feed_forward.w1`.

Overall the reason for learning rate transfer without weight decay, to the extent it occurs, would then be similar as in the independent weight decay case. Early in training $\mu$P correctly scales down the relative updates which helps transfer relative representation changes under the high, width-dependent, initial alignment ratios. Later in training the relative updates become approximately the same across widths which helps stabilize the rate of feature learning when the alignment ratios are identical across widths. Only standard weight decay follows $\mu$P's prescribed relative weight update scaling from Equation 6 throughout training, which is exactly why learning rates fail to transfer with standard weight decay. With either no weight decay or independent weight decay, the relative updates deviate from the prescribed scaling which helps counteract the violated alignment assumptions. However, independent weight decay seems to control the relative updates later in training more tightly, resulting in better transfer overall.

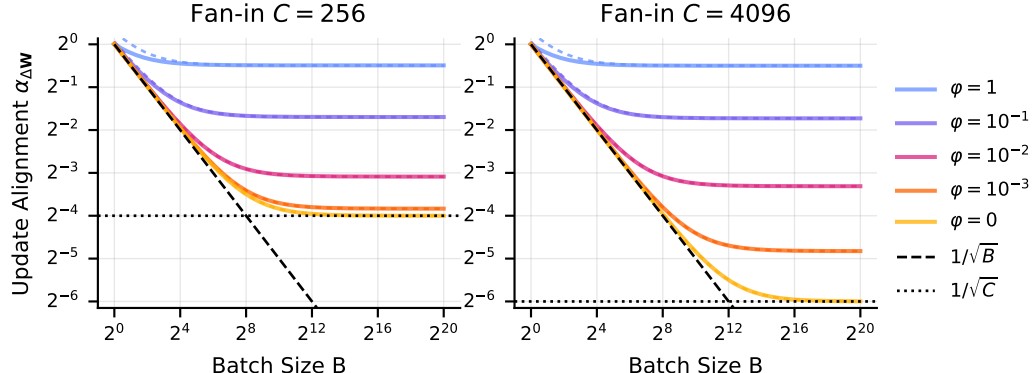

**Figure 10: Predicted update alignment $\alpha_{\Delta W}$ in the simple noise model based on Equation 25.**
The overall behavior is well approximated as $\alpha_{\Delta W} \approx \sqrt{C^{-1} + B^{-1} + \varphi/(1+\varphi)}$ shown by the dashed color lines. Note how the alignment for a low signal-to-noise ratio $\varphi \approx 0$ is roughly $1/\sqrt{B}$ (no width-dependence as $\mu$P assumes) until $B > C$ when it becomes $1/\sqrt{C}$ ($\mu$P's assumption fails). Higher values of $\varphi$ always increase the alignment regardless of $B$ and $C$. This is most prominent when $\varphi \gg 1/C$ in which case we get $\alpha_{\Delta W} \approx 1/\sqrt{\varphi^{-1}+1}$ once $B \gg \varphi^{-1}$. This can be viewed as the batch size being effectively capped at $\varphi^{-1}+1$, an estimate of the critical batch size (McCandlish et al., 2018). Early in training this can give a high approximately width-independent alignment, allowing $\mu$P's assumptions to hold initially even when $B \gg C$ but then fail as the SNR decreases.

## C  ANALYSIS OF THE UPDATE ALIGNMENT FOR A SIMPLE NOISE MODEL

In this section we analyze the update alignment for a simple probabilistic model of a single neuron with a fan-in of $C$ at batch size $B$. On the forward pass we have:

$$\mathbf{y}^\top = [\mathbf{y}_1, \ldots, \mathbf{y}_B] = \mathbf{w}^\top \mathbf{X} = \mathbf{w}^\top [\mathbf{x}_1, \ldots, \mathbf{x}_B] \tag{14}$$

where $\mathbf{w} \in \mathbb{R}^C$ are the weights of the neuron, $\mathbf{X} \in \mathbb{R}^{C \times B}$ is a batch of $B$ input vectors with dimension $C$ each, and $\mathbf{y} \in \mathbb{R}^B$ is the output for each input. The gradient of the final average sample loss $\mathscr{L} = \frac{1}{B} \sum_{b=1}^B \mathscr{L}_b$ with respect to the weights $\mathbf{w}$ is given by:

$$\mathbf{g} := \frac{\partial \mathscr{L}}{\partial \mathbf{w}} = \frac{1}{B} \sum_{b=1}^B \frac{\partial \mathscr{L}_b}{\partial \mathbf{y}_b} \mathbf{x}_b = \frac{1}{B} \sum_{b=1}^B \mathbf{y}_b' \mathbf{x}_b \tag{15}$$

where we have defined $\mathbf{y}_b' := \partial \mathscr{L}_b / \partial \mathbf{y}_b$. Note how the gradient for each sample $\mathbf{g}_b := \mathbf{y}_b' \mathbf{x}_b$ is simply a scaled version of the input vector $\mathbf{x}_b$ and is thus fully aligned with it.

We want to be able to introduce analytically tractable correlations between the random variables in our probabilistic model. Taking inspiration from Kosson et al. (2024a), we use the Signal-to-Noise ratio $\varphi$ to capture the correlation. It is defined as the square ratio of the norms of the expected gradient (signal) to the expected deviation (noise):

$$\varphi := \frac{\|\bar{\mathbf{g}}\|^2}{\mathbb{E}_b[\|\mathbf{g}_b - \bar{\mathbf{g}}\|^2]}, \qquad \bar{\mathbf{g}} := \mathbb{E}_b[\mathbf{g}_b] \tag{16}$$

We can now define a distributions for $\mathbf{x}_b$ and $\mathbf{y}_b'$ that result in a specific $\varphi$ while respecting the mathematical form of the gradient and being simple enough to analyze. We chose the form of the input samples as:

$$\mathbf{x}_b = (\tilde{\mathbf{x}}_b + \sqrt{\varphi} \cdot \mathbf{s} \cdot \frac{1}{\mathbf{y}_b'}) / \sqrt{1 + \varphi} \tag{17}$$

where the noise component $\tilde{\mathbf{x}}_b$ and signal component $\mathbf{s}$ are independent random vectors drawn from a standard multivariate normal distribution, i.e., $\tilde{\mathbf{x}}_b, \mathbf{s} \sim \mathcal{N}(\mathbf{0}, I_C)$. We choose the distribution of the output gradient to be $\mathbf{y}_b' = \pm 1$ with equal probability to avoid complexities in modeling the inverse.

We could scale the variance of the distributions but this would not affect the alignment ratio. In this model the gradient becomes:

$$\mathbf{g}_b = \mathbf{y}'_b\mathbf{x}_b = (\tilde{\mathbf{x}}_b\mathbf{y}'_b + \sqrt{\varphi}\cdot\mathbf{s})/\sqrt{1+\varphi} \tag{18}$$

Assuming $\|\mathbf{s}\|^2 = C$ (or alternatively in expectation over $\mathbf{s}$), this results in the targeted signal-to-noise ratio $\varphi$:

$$\frac{\|\bar{\mathbf{g}}\|^2}{\mathbb{E}_b[\|\mathbf{g}_b - \bar{\mathbf{g}}\|^2]} = \frac{\varphi\|\mathbf{s}\|^2/(1+\varphi)}{\mathbb{E}_b[\|\mathbf{y}'_b\tilde{\mathbf{x}}_b/\sqrt{1+\varphi}\|^2]} = \varphi\frac{C/(1+\varphi)}{C/(1+\varphi)} = \varphi \tag{19}$$

Note that there are other forms for $\mathbf{x}_b$ that result in the desired signal-to-noise ratio including some that give $\bar{\mathbf{g}} = \mathbf{s}$, but our choice results in bounded input and gradient norms in expectation for both $\varphi \to 0$ and $\varphi \to \infty$. We will also assume that the update is obtained via simple gradient descent:

$$\Delta\mathbf{w} = -\eta\mathbf{g} \tag{20}$$

The update alignment involves a ratio that is hard to analyze under expectation. Instead we will approximate this as the ratio of the square expectations:

$$\mathbb{E}[\alpha_{\Delta W}] = \mathbb{E}\left[\frac{\|\Delta\mathbf{w}^\top\mathbf{X}\|}{\|\Delta\mathbf{w}\|\|\mathbf{X}\|}\right] \approx \sqrt{\frac{\mathbb{E}[\|\Delta\mathbf{w}^\top\mathbf{X}\|^2]}{\mathbb{E}[\|\Delta\mathbf{w}\|^2]\mathbb{E}[\|\mathbf{X}\|^2]}} \tag{21}$$

we thus need to compute $\mathbb{E}[\|\Delta\mathbf{w}^\top\mathbf{X}\|^2]$, $\mathbb{E}[\|\Delta\mathbf{w}\|^2]$, and $E[\|\mathbf{X}\|^2]$.

Starting with the input size we get:

$$\mathbb{E}[\|\mathbf{X}\|^2] = B \cdot \mathbb{E}_b[\|\mathbf{x}_b\|^2] \tag{22a}$$

$$= B \cdot \mathbb{E}_b\left[\frac{1}{1+\varphi}\left\langle\tilde{\mathbf{x}}_b + \sqrt{\varphi}\cdot\mathbf{s}\cdot\frac{1}{\mathbf{y}'_b}, \tilde{\mathbf{x}}_b + \sqrt{\varphi}\cdot\mathbf{s}\cdot\frac{1}{\mathbf{y}'_b}\right\rangle\right] \tag{22b}$$

$$= \frac{B}{1+\varphi}\left(\mathbb{E}_b[\|\tilde{\mathbf{x}}_b\|^2] + 2\sqrt{\varphi}\mathbb{E}_b\left[\frac{1}{\mathbf{y}'_b}\langle\tilde{\mathbf{x}}_b, \mathbf{s}\rangle\right] + \varphi\mathbb{E}_b\left[\frac{1}{(\mathbf{y}'_b)^2}\|\mathbf{s}\|^2\right]\right) \tag{22c}$$

$$= \frac{B}{1+\varphi}(C + 0 + \varphi\cdot C) \tag{22d}$$

$$= BC \tag{22e}$$

For the update size we get:

$$\mathbb{E}[\|\Delta\mathbf{w}\|^2] = \eta^2\mathbb{E}[\|\mathbf{g}\|^2] \tag{23a}$$

$$= \eta^2\mathbb{E}\left[\left\|\frac{1}{B}\sum_{b=1}^B\frac{\mathbf{y}'_b\tilde{\mathbf{x}}_b + \sqrt{\varphi}\mathbf{s}}{\sqrt{1+\varphi}}\right\|^2\right] \tag{23b}$$

$$= \frac{\eta^2}{B^2(1+\varphi)}\mathbb{E}\left[\left\|\left(\sum_{b=1}^B\mathbf{y}'_b\tilde{\mathbf{x}}_b\right) + B\sqrt{\varphi}\mathbf{s}\right\|^2\right] \tag{23c}$$

$$= \frac{\eta^2}{B^2(1+\varphi)}\left(\mathbb{E}\left[\left\|\sum_{b=1}^B\mathbf{y}'_b\tilde{\mathbf{x}}_b\right\|^2\right] + 2\mathbb{E}\left[\left\langle\sum_{b=1}^B\mathbf{y}'_b\tilde{\mathbf{x}}_b, B\sqrt{\varphi}\mathbf{s}\right\rangle\right] + \mathbb{E}\left[\|B\sqrt{\varphi}\mathbf{s}\|^2\right]\right) \tag{23d}$$

$$= \frac{\eta^2}{B^2(1+\varphi)}\left(\sum_{b=1}^B\mathbb{E}[\|\tilde{\mathbf{x}}_b\|^2] + 0 + B^2\varphi\mathbb{E}[\|\mathbf{s}\|^2]\right) \tag{23e}$$

$$= \frac{\eta^2}{B^2(1+\varphi)}(BC + B^2\varphi C) \tag{23f}$$

$$= \frac{\eta^2C}{B}\frac{1+B\varphi}{1+\varphi} \tag{23g}$$

Finally we can compute the expected output change as:

$$\mathbb{E}[\|\Delta\mathbf{w}\mathbf{X}\|^2] = B\mathbb{E}[\langle\mathbf{x}_b, -\eta\mathbf{g}\rangle^2] \tag{24a}$$

$$= \frac{B\eta^2}{(1+\varphi)^2}\mathbb{E}\Big[\Big\langle\tilde{\mathbf{x}}_b + \sqrt{\varphi}\mathbf{s}/\mathrm{y}_b', \frac{1}{B}\sum_{i=1}^{B}(\tilde{\mathbf{x}}_i\mathrm{y}_i' + \sqrt{\varphi}\mathbf{s})\Big\rangle^2\Big] \tag{24b}$$

$$= \frac{\eta^2}{(1+\varphi)^2 B}\mathbb{E}\Big[\Big(\mathrm{y}_b'\|\tilde{\mathbf{x}}_b\|^2 + \sum_{i\neq b}\mathrm{y}_i'\langle\tilde{\mathbf{x}}_b, \tilde{\mathbf{x}}_i\rangle + (1+B)\sqrt{\varphi}\langle\tilde{\mathbf{x}}_b, \mathbf{s}\rangle$$
$$+ \frac{\sqrt{\varphi}}{\mathrm{y}_b'}\sum_{i\neq b}\langle\mathbf{s}, \tilde{\mathbf{x}}_i\rangle\mathrm{y}_i' + \frac{B\varphi}{\mathrm{y}_b'}\|\mathbf{s}\|^2\Big)^2\Big] \tag{24c}$$

$$= \frac{\eta^2}{(1+\varphi)^2 B}\Big[\mathbb{E}[\|\tilde{\mathbf{x}}_b\|^4] + \mathbb{E}\Big[\Big(\sum_{i\neq b}\mathrm{y}_i'\langle\tilde{\mathbf{x}}_b, \tilde{\mathbf{x}}_i\rangle\Big)^2\Big] + (1+B)^2\varphi\mathbb{E}[\langle\tilde{\mathbf{x}}_b, \mathbf{s}\rangle^2]$$
$$+ \varphi\mathbb{E}\Big[\Big(\sum_{i\neq b}\langle\mathbf{s}, \tilde{\mathbf{x}}_i\rangle\mathrm{y}_i'\Big)^2\Big] + B^2\varphi^2\mathbb{E}\Big[\|\mathbf{s}\|^4\Big] + 2B\varphi\mathbb{E}\Big[\|\tilde{\mathbf{x}}_b\|^2\|\mathbf{s}\|^2\Big]\Big] \tag{24d}$$

$$= \frac{\eta^2}{(1+\varphi)^2 B}\Big[\mathbb{E}[\|\tilde{\mathbf{x}}_b\|^4] + \sum_{i\neq b}\mathbb{E}[\langle\tilde{\mathbf{x}}_b, \tilde{\mathbf{x}}_i\rangle^2] + (1+B)^2\varphi\mathbb{E}[\langle\tilde{\mathbf{x}}_b, \mathbf{s}\rangle^2]$$
$$+ \varphi\sum_{i\neq b}\mathbb{E}[\langle\mathbf{s}, \tilde{\mathbf{x}}_i\rangle^2] + B^2\varphi^2\mathbb{E}[\|\mathbf{s}\|^4] + 2B\varphi\mathbb{E}[\|\tilde{\mathbf{x}}_b\|^2\|\mathbf{s}\|^2]\Big] \tag{24e}$$

$$= \frac{\eta^2}{(1+\varphi)^2 B}\Big[C(C+2) + (B-1)C + (1+B)^2\varphi C$$
$$+ \varphi(B-1)C + B^2\varphi^2 C(C+2) + 2B\varphi C^2\Big] \tag{24f}$$

Now we have all the factors needed to approximate the update alignment via Equation 21 giving:

$$\mathbb{E}[\alpha_{\Delta\mathbf{W}}] \approx \sqrt{\frac{(C+2) + (B-1) + (1+B)^2\varphi + \varphi(B-1) + B^2\varphi^2(C+2) + 2B\varphi C}{(1+B\varphi)(1+\varphi)CB}} \tag{25}$$

Figure 10 plots this prediction showing $C$-dependent behavior for large batch-to-width ratios as expected. We find the overall behavior in Equation 25 well approximated by:

$$\mathbb{E}[\alpha_{\Delta\mathbf{W}}] \approx \sqrt{C^{-1} + B^{-1} + \varphi/(1+\varphi)} \tag{26}$$

The last term is the inverse of $\varphi^{-1}+1$ which is an estimate of the critical batch size (McCandlish et al., 2018) where the overall gradient starts to become dominated by the signal component and the benefit of increasing the batch size further diminishes quickly. In the fully uncorrelated case $\varphi = 0$ we roughly get $\alpha_{\Delta\mathbf{W}} \approx 1/\sqrt{B}$ when $B \ll C$ which is the regime targeted by $\mu$P, infinitely wide networks trained with finite batch sizes. In the practical training settings we often have $B \gg C$ which results in $\alpha_{\Delta\mathbf{W}} \approx 1/\sqrt{C}$. This $C$ dependence breaks the scaling behavior of $\mu$P. When the critical batch size is small compared to $C$ this can instead become $\alpha_{\Delta\mathbf{W}} \approx \sqrt{\frac{\varphi}{1+\varphi}}$ which eliminates the $C$ dependence again. The overall alignment behavior is thus highly dependent on the specific settings of $B$ and $C$ as well as the signal-to-noise ratio $\varphi$ which dynamically changes throughout training.

# D ANALYZING THE SCHEDULING EFFECTS OF WEIGHT DECAY

## D.1 THE EVOLUTION OF THE WEIGHT NORM OVER TIME

Here we construct a simple model of how the weight norms evolve over time, similar to the one used in prior work (Kosson et al., 2024b). In this section we use $\Delta \boldsymbol{W}$ to refer to this "gradient component" of the update, treating the weight decay component separately. We will assume that the weight update (excluding weight decay) at each timestep is orthogonal to the current weights. This results in a simple recurrence relation for the weight norms:

$$\|\mathbf{W}_{t+1}\|^2 = (1 - \eta_t \lambda)^2 \|\mathbf{W}_t\|^2 + \|\Delta \boldsymbol{W}_t\|^2 \tag{27}$$

We will assume that the elementwise RMS size of the weight update is roughly one, for example due to the second moment normalization in Adam. Defining $\rho_t := \|\boldsymbol{W}_t\|/\sqrt{KC}$ as the RMS value of the elements in $\|\boldsymbol{W}_t\|$ and $a_t := 1 - \eta_t \lambda$ as the decay multiplier we get a recurrence relation:

$$\rho_{t+1}^2 = a_t^2 \rho_t^2 + \eta_t^2 \tag{28}$$

This model involves several significant simplifications:

- The assumption that the update is orthogonal to the current weights can either come from randomness or due to scale-invariance from normalization layers. The gradient of a scale-invariant weight is always orthogonal to the weight and zero-mean uncorrelated random gradient is orthogonal to the weights on average. We note that with momentum the update is not necessarily orthogonal on average, even in these special cases.

- This model only holds for momentum when the gradients are uncorrelated over time. In this case we can replace the update term by the total contribution of the current gradient over time (the sum of the impulse response) as done by Kosson et al. (2024b). This also assumes $\beta_1$ is not too large, so that the impact of a single gradient happens over a short timeframe compared to the $t$ values we are interested in. If there is correlation the behavior of the system changes in a way we can not predict without additional information. See Wan et al. (2021) for an analysis that partially accounts for this in SGDM.

- The size of the update assumes that $\beta_2$ can accurately track the expected magnitude of the gradient over time. This requires sufficiently small $\beta_2$ values in comparison to how fast the gradient magnitude changes. We note that at the start of training the gradient magnitude often changes very rapidly.

This means that in practice Equation 28 does not accurately predict the behavior of the weight norm in neural network training. However, we believe it is the best we can do based on the hyperparameters alone without measurements of the temporal gradient correlation and gradient magnitude over time (which we expect strongly depend on the other hyperparameters too). We still find the predictions of this model to be informative for real training, particularly those based on the equilibrium value that satisfies $\rho_{t+1} = \rho_t$. The RMS values seen in real training often stabilize near this value (Kosson et al., 2024b), but how fast this happens depends strongly on the complex temporal effects described above.

With this disclaimer, we proceed with the analysis of Equation 28. Rolling out the recurrence relation:

$$\rho_{t+1}^2 = \rho_0^2 \prod_{\tau=0}^t a_\tau^2 + \sum_{\tau=0}^t \eta_\tau^2 \prod_{i=\tau+1}^t a_i^2 \tag{29}$$

In the case where the learning rate is constant $\eta_t = \eta$, making $a_t = a$ constant as well, we get:

$$\rho_t^2 = \rho_0^2 \cdot a^{2t} + \eta^2 \sum_{\tau=0}^{t-1} a^{2(t-\tau)} = \rho_0^2 \cdot a^{2t} + \eta^2 \frac{1 - a^{2t}}{1 - a^2} \tag{30}$$

assuming $0 < a < 1$. If $\eta_t$ is not constant, we can still easily compute the behavior of the weight norm over time using the recurrence relation or the rolled-out form directly.

In the constant learning rate case the RMS value approaches the equilibrium value $\rho_\infty$ exponentially:

$$\rho_t^2 = \rho_\infty^2 + (\rho_0^2 - \rho_\infty^2)a^{2t} \qquad\qquad \rho_\infty^2 = \frac{\eta^2}{1 - a^2} \approx \frac{\eta}{2\lambda} \tag{31}$$

We can use this to estimate how long it takes for the weights to approximately reach equilibrium. If we define this by the timestep $T_{EQ}$ when the RMS value falls within a factor $\sqrt{2}$ of the equilibrium value we get:

$$
T_{\text{EQ}} \approx
\begin{cases}
\frac{1}{2\eta\lambda} \ln\left(\frac{1-\rho_0^2/\rho_\infty^2}{1-1/\sqrt{2}}\right) & \text{if } \rho_0 < \frac{1}{\sqrt{2}}\rho_\infty \\
\frac{1}{2\eta\lambda} \ln\left(\frac{\rho_0^2/\rho_\infty^2-1}{\sqrt{2}-1}\right) & \text{if } \rho_0 > \sqrt{2}\rho_\infty \\
0 & \text{otherwise}
\end{cases}
\tag{32}
$$

### D.2 THE EVOLUTION OF THE RELATIVE WEIGHT UPDATE OVER TIME

In this simplified model the relative weight update becomes:

$$
\frac{\|\Delta \boldsymbol{W}_t\|}{\|\boldsymbol{W}\|} = \frac{\eta_t}{\rho_t}
\tag{33}
$$

which we can compute exactly given the expressions for $\rho_t$ from the previous subsection.

In the constant learning rate case we get:

$$
\frac{\|\Delta \boldsymbol{W}_t\|}{\|\boldsymbol{W}\|} = \sqrt{\frac{\eta^2}{\rho_t^2}} = \sqrt{\frac{\eta^2}{\rho_0^2 \cdot a^{2t} + \eta^2 \frac{1-a^{2t}}{1-a^2}}} \xrightarrow[t\to\infty]{} \sqrt{2\eta\lambda - \eta^2\lambda^2} \approx \sqrt{2\eta\lambda}
\tag{34}
$$

showing that the product $\eta\lambda$ determines the size of relative weight updates in the steady-state. However, at the start of training, e.g. the first step, the relative weight update (excluding the weight decay component) is determined by just the learning rate and the initialization norm.

It is also interesting to see what happens without weight decay, $\lambda = 0$. For a constant learning rate, this gives:

$$
\rho_t^2 = \rho_0^2 + t \cdot \eta^2 \qquad\qquad \frac{\eta_t}{\rho_t} = \frac{\eta}{\sqrt{\rho_0^2 + t \cdot \eta^2}} = \frac{1}{\sqrt{(\rho_0^2/\eta^2) + t}}
\tag{35}
$$

which means that the relative updates decay towards zero over time roughly as $t^{-0.5}$, rather than stabilizing at a fixed value. For this reason it is important to use learning rate schedules with weight decay, otherwise the effective learning rate measured in terms of relative updates never decays which typically leads to suboptimal outcomes. This decay in the relative update size can also be detrimental, if the relative update size decreases too quickly the network stops learning (sometimes referred to as a loss of plasticity). An example of this was seen in the AdamW paper (Loshchilov & Hutter, 2019) where the optimal weight decay value in an experiment was zero for a constant learning rate but non-zero when a cosine schedule was used. It is better to vary the effective learning rate over time explicitly according to our desired schedule, ideally exactly through optimizers like LionAR (Kosson et al., 2024a) or approximately via weight decay.

### D.3 THE WARMUP EFFECT OF HIGH WEIGHT DECAY CONFIGURATIONS

Let us now consider two configurations for the learning rate and weight decay, $(\eta, \lambda)$ vs $(\eta/m, m\lambda)$. These correspond to the configurations from independent weight decay scaling compared to no scaling (or in standard scaling at a higher base learning rate). We first note that their product is identical, so the relative weight update will converge to the same equilibrium value. However, the very first update will also be $m$ times smaller for the high weight decay configuration. A learning rate warmup starting at $\eta/m$ and reaching $\eta$ over time has an identical effect on the initial update and the steady-state updates. However, the exact shape of this warmup effect is not trivial, i.e. how it affects the relative update size over time.

We use the previous symbols to denote quantities for the base hyperparameters $(\eta, \lambda)$ and $\hat{\phantom{x}}$ over the symbol to denote the corresponding quantities for the scaled configuration. We are interested in the ratio of the relative weight updates between the two configurations:

$$
s_t := \frac{\widehat{\eta}_t/\widehat{\rho}_t}{\eta_t/\rho_t}
\tag{36}
$$

which captures the size of the warmup effect. For the simple weight norm model we can always compute this using Equation 28, even when the learning rate varies over time. For constant learning rates we would get:

$$s_t = \frac{1}{m} \frac{\rho_t}{\widehat{\rho_t}} \tag{37a}$$

$$= \frac{1}{m} \sqrt{\frac{\rho_0^2 \cdot a^{2t} + b^2 \eta^2 \frac{1-a^{2t}}{1-a^2}}{\rho_0^2 \cdot a^{2t} + b^2 \widehat{\eta}^2 \frac{1-a^{2t}}{1-a^2}}} \tag{37b}$$

$$= \sqrt{\frac{\rho_\infty^2 + (\rho_0^2 - \rho_\infty^2)a^{2t}}{\rho_\infty^2 + (m^2\rho_0^2 - \rho_\infty^2)a^{2t}}} \tag{37c}$$

$$= \sqrt{\frac{1 + (\rho_0^2/\rho_\infty^2 - 1)a^{2t}}{1 + (m^2\rho_0^2/\rho_\infty^2 - 1)a^{2t}}} \tag{37d}$$

This is already a relatively complicated expression with behavior that depends on the initialization magnitude $\rho_0$ as well as the exact values of $\eta$ and $\lambda$ used. The effective length of the additional warmup can vary from a single step, e.g. when $\rho_0 \to 0$, to the whole training, for example when $\rho_0$ is large or decays slowly. We note it will also depend significantly on $\eta\lambda$, so when sweeping learning rates the length of the warmup effect will differ between points.

Finally we note that simply scaling the learning rate schedule for the base configuration by $s_t$ will generally not give relative updates matching the high weight decay configuration. This is because modifying the learning rate schedule also changes how the norms evolve over time. A scaling factor that would make the profiles match exactly could still be computed using the recurrence relations by keeping track of how the modifications affect the norms over time. However, exactly matching the warmup effect from higher weight decay values may not be that important since it is largely arbitrary as far as we know in the sense that it does not depend on the alignment ratio. Applying the scaling factor from Equation 36 to a relative or rotational optimizer like LionAR (Kosson et al., 2024a) would result in the desired warmup effect.

## E  RESNET EXPERIMENTS

In this section we repeat some of our experiments for ResNet training to see if they generalize to other architectures and settings. The training setup is described in Appendix F.4. Overall our main conclusions from the LLaMA experiments hold well. The main differences are that the learning rate transfer is worse, likely due to regularization effects and that the warmup effects seem to matter less. The results are shown in the following figures.

- Figure 11 compares learning rate transfer between standard and independent weight decay scaling. Independent weight decay gives better transfer but there is a shift in the optimal learning rates for both scaling approaches.

- Figure 12 shows that independent weight decay scaling helps preserve relative representation changes. The behavior is essentially identical to the LLaMA setting.

- Figure 13 measures the weight alignment, update alignment and alignment ratio. Just like in the LLaMA setting $\mu$P's assumptions only hold for a very brief period at the start of training. The weight alignment changes more than we observed with LLaMA.

- Figure 14 empirically measures the warmup effect from the use of high weight decay. The conclusions are identical to the LLaMA setting.

- Figure 15 compares learning rate transfer between $\mu$P with independent learning rate scaling and not applying $\mu$P scaling. We find that both learning rate transfer quality and stability are similar. The $\mu$P configuration results in a slightly higher accuracy which may at least in part be due to the warmup effect.

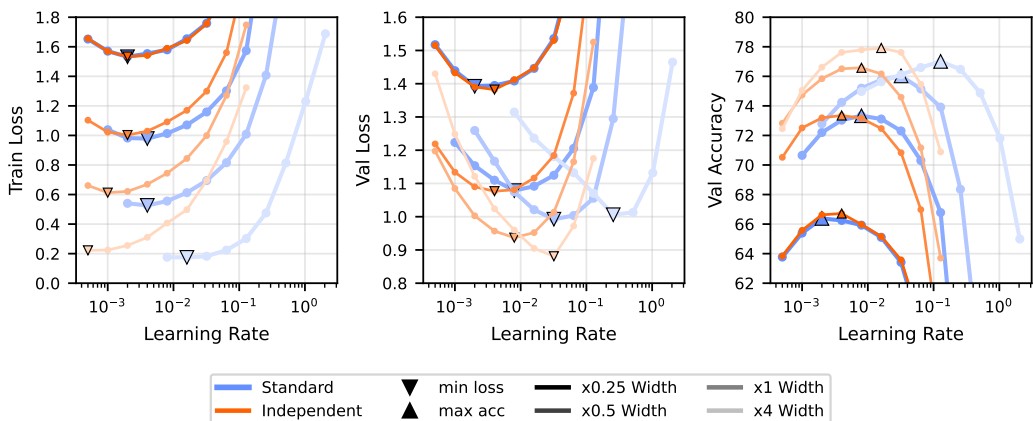

**Figure 11: Comparison of standard vs independent $\mu$P scaling for ResNet training on ImageNet.**
Learning rates generally transfer much worse than for LLaMA. Independent scaling still gives better results than standard scaling overall. Note how optimal learning rates in terms of training loss shift downwards with independent scaling while they shift upwards for the validation loss and accuracy, a strong indication of a confounding regularization effect. Widths are specified relative to a standard ResNet-50 but the base width for scaling is the $0.25\times$ variant. Learning rates for parameters that do not scale with width are frozen before the sweep at the optimal value for the $0.25\times$ network.

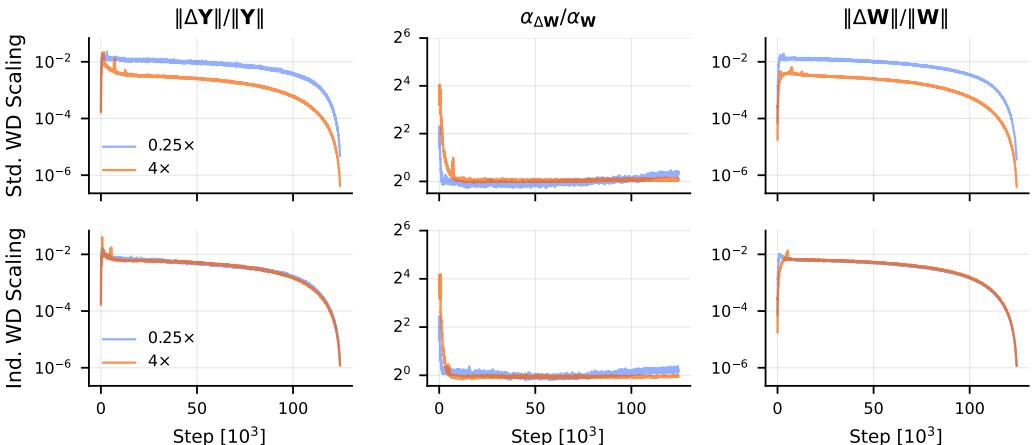

**Figure 12:** ResNet variant of Figure 3. Similar to the LLaMA case, standard weight decay scaling fails to preserve the relative representation change across widths. The alignment ratio is similar across widths contrary to $\mu$P's assumptions, requiring maintaining relative weight updates as achieved by independent weight decay scaling. Plots show for `layer3.2.conv1` which is a $1 \times 1$ convolutional layer with `fan_in=4096` for the $4\times$ variant. The learning rate used here is $\eta = 4 \times 10^{-3}$ with a 1 epoch warmup (1%) followed by cosine decay to zero.

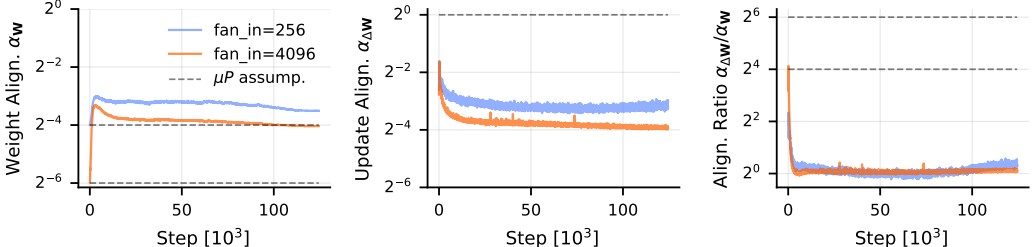

**Figure 13:** ResNet variant of Figure 4, showing how the alignment assumptions of $\mu$P do not hold. Plots show for `layer3.2.conv1` which is a $1 \times 1$ convolutional layer. Both the weight alignment and update alignment deviate significantly from $\mu$P's assumptions. Note the larger changes in the weight alignment over time compared to the LLaMA experiment. Learning rate $\eta = 4 \times 10^{-3}$, no $\mu$P scaling applied.

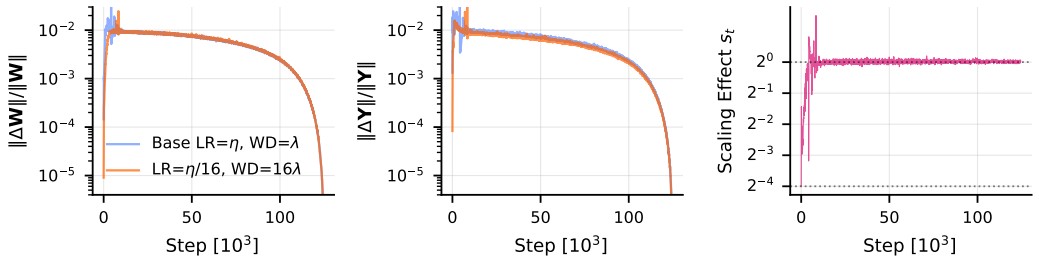

**Figure 14:** ResNet version of Figure 5, showing the warmup effect of higher weight decay configurations. Plots show for `layer3.2.conv1` which is a $1 \times 1$ convolutional layer. Hyperparameters used are $\eta = 8 \times 10^{-3}$, $\lambda = 0.1$ corresponding to the best standard WD configuration in Figure 11 (after $\mu$P scaling). The additional warmup effect here is comparable in absolute length to the LLaMA setting but proportionally shorter.

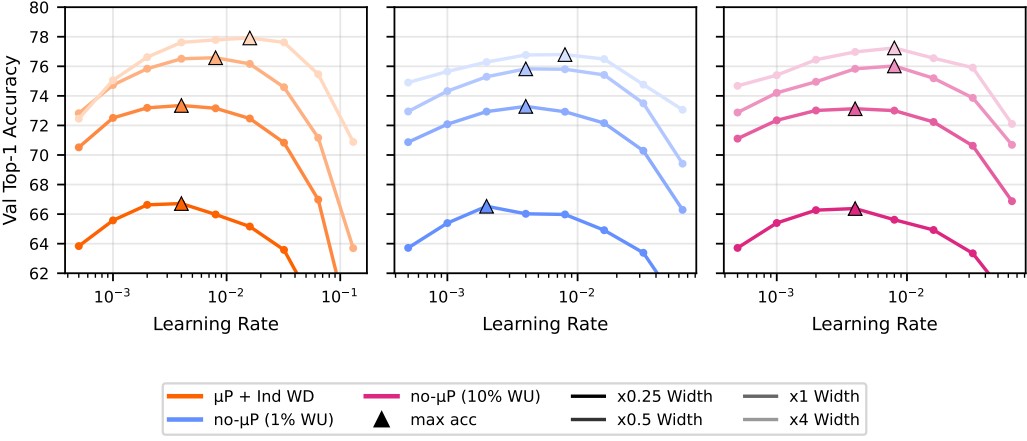

**Figure 15:** Learning rate transfer comparison between $\mu$P with independent weight decay scaling (left) and no-$\mu$P scaling for ResNet training (middle/right). The left and middle panels use a 1% linear warmup, the right panel uses a 10% linear warmup (WU). The additional warmup effect of $\mu$P with independent weight decay scaling is not needed for stability or transfer, but may contribute to the slightly higher accuracy achieved on the left. Other differences such as the resulting weight norms could also matter, particularly for the final FC layer. For normalized layers the weight norms also inversely scale the gradient norms which may affect the result through interactions with the epsilon value of Adam. A similar gap also appears in Figure 11 between standard and independent weight decay scaling.

---

**Algorithm 1** The PyTorch variant of AdamW. This differs from the original by Loshchilov & Hutter (2019) which corresponds to replacing the orange term with $\eta_t / \max_t(\eta_t)$ here, applying the learning rate schedule to the weight decay term but decoupling the overall strength from the peak LR $\max_t(\eta_t)$.

---

**Require:** Learning rate $\eta_t$, weight decay $\lambda$, momentum $\beta_1$, magnitude smoothing $\beta_2$, $\epsilon$ for numerical stability
1: **Initialize:** Time step $t \leftarrow 0$, parameter vector $\boldsymbol{\theta}_0$, momentum vector $\boldsymbol{m}_0 \leftarrow 0$, magnitude vector $\boldsymbol{v}_0 \leftarrow 0$
2: **while** stopping criteria not met **:**
3: $\quad t \leftarrow t + 1$
4: $\quad \boldsymbol{g}_t \leftarrow$ Mini-batch gradient w.r.t. $\boldsymbol{\theta}_{t-1}$
5: $\quad \boldsymbol{m}_t \leftarrow \beta_1 \boldsymbol{m}_{t-1} + (1 - \beta_1)\boldsymbol{g}_t$
6: $\quad \boldsymbol{v}_t \leftarrow \beta_2 \boldsymbol{v}_{t-1} + (1 - \beta_2)\boldsymbol{g}_t^2$
7: $\quad \hat{m}_t \leftarrow \boldsymbol{m}_t / (1 - \beta_1^t)$
8: $\quad \hat{v}_t \leftarrow \boldsymbol{v}_t / (1 - \beta_2^t)$
9: $\quad \boldsymbol{\theta}_t \leftarrow (1 - \eta_t \lambda)\boldsymbol{\theta}_{t-1} - \eta_t \hat{m}_t / (\sqrt{\hat{v}_t} + \epsilon)$

---

# F    EXPERIMENTAL DETAILS

## F.1    ALGORITHMS

The AdamW variant we use is shown in Algorithm 1. Note how the weight decay hyperparameter in line 9 is scaled by the learning rate. This differs from the original variant described in Loshchilov & Hutter (2019), which is often referred to as having "independent" or "fully-decoupled" weight decay. This variant only has $\lambda$ instead of $\eta\lambda$ in line 9, but the weight decay hyperparameter is still varied according to the same schedule as the learning rate.

## F.2    LEARNING RATE TRANSFER EXPERIMENTS

Our learning rate transfer experiments consist of two phases. We start from an existing hyperparameter configuration and first sweep the global learning rate used across all parameter types, i.e., the input layer, hidden layers, output layer, biases and gains. This is done for the $1\times$ scale model. We then freeze the learning rate of the input layer, biases and gains, changing only the learning rate for the layers that scale with the width, i.e., the hidden layer and output layer. This second base learning rate is scaled via $\mu$P's rules before applying it in the optimizer, but the plots show the base rate before scaling (which is what transfers across scale). This second sweep gives the final performance shown in the learning rate transfer plots.

We adopt this approach because it better captures how well the learning rate scaling works by leaving the parameters it is not applied to unaffected. Compared to sweeping a single global learning rate across scales, this typically gives better performance with standard scaling but highlights the shift more. The reason is that without freezing the learning rate for the other layers, they can become unstable at the shifted (higher) base learning rates needed for the hidden layers to learn well. This gives worse overall results since no single global learning rate works well for all parameters anymore, but the optimal learning rates may shift less as a result.

Instead of using the full Maximal Update Parameterization $\mu$P for training we combine standard parameterization with the learning rate scaling of $\mu$P as proposed by Everett et al. (2024). They find this simpler approach empirically works just as well or better. We note there are many variants of $\mu$P that achieve the same effects slightly differently, see for example Tables 3, 8 and 9 in Tensor Programs V (Yang et al., 2021). However, these are generally not equivalent with standard weight decay, which would complicate the use of full $\mu$P for our experiments.

For transformer training the main differences between full $\mu$P and our experiments are the handling of the final output layer and the normalization factor applied in attention. The output layer is initialized using a smaller standard deviation that scales with $1/C$ rather than $1/\sqrt{C}$ like in standard parameterization. This results in an initial relative update size that is not scaled with the width compared to the $1/\sqrt{m}$ scaling that occurs in the approach from Everett et al. (2024). The other difference is that the scaling factor in the attention heads is $1/d_{\text{head}}$ rather than $1/\sqrt{d_{\text{head}}}$. However we keep $d_{\text{head}}$ constant in our scaling experiments, opting to scale the number of heads instead. This difference could thus be absorbed into a tunable factor that does not change across scales.

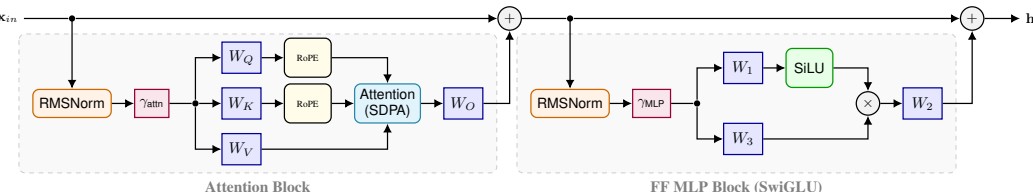

**Figure 16: LLaMA transformer block diagram**. The names of the matrices are used to refer to measurements in other figures. The learnable gains $\gamma_{\text{attn}}$ and $\gamma_{\text{MLP}}$ are $d_{\text{emb}}$-dimensional vectors.

### F.3 LLaMA EXPERIMENT CONFIGURATION

Our LLaMA Touvron et al. (2023) experiments are performed using a modified version of Lingua (Videau et al., 2024). The networks have 20 transformer blocks, each consisting of one attention and one MLP sub-block (see Figure 16). We vary the embedding dimension $d_{\text{emb}}$ between 128 and 2048, which we sometimes refer to as $1\times$ and $16\times$. The MLP block has an expansion factor of exactly $3\times$ giving exact scaling ratios for all layers. We used a fixed attention head size of $d_{\text{head}} = 128$, varying the number of heads across scale. All linear layers are initialized using a standard deviation of $1/\sqrt{C}$. We do not use embedding tying but the embeddings are still initialized with a standard deviation of $1/\sqrt{d_{\text{emb}}}$ like the final FC layer (default behavior in Lingua).

The task is standard next token prediction pre-training using a cross-entropy loss. The dataset used is DCLM (Li et al., 2024), specifically a 10% subset obtained by using the first global shard from the Hugging Face dataset. We use the llama3 tokenizer[3] with a vocabulary size of 128,256.

Unless otherwise stated we use the PyTorch variant of the AdamW optimizer shown in Algorithm 1. We vary the learning rate between runs but other hyperparameters are fixed at $\beta_1 = 0.9$, $\beta_2 = 0.95$, $\epsilon = 10^{-8}$, and $\lambda = 0.1$ (except when scaled through independent weight decay scaling). Weight decay is applied to all parameters except the gains. We train for 20,000 steps at a global batch size of 256 and sequence length of 4096 giving 1,048,576 ($2^{20}$) tokens in each batch and 20.97 billion tokens in total. This results in 19.23 tokens per non-embedding parameter in the largest configuration we use (embedding dim 2048). Unless otherwise noted, the learning rate schedule consists of a 10% linear warmup from 0 to the specified peak learning rate, followed by a linear decay down to 0. The losses reported are the average of the last 100 training steps unless otherwise specified. This is due to initial issues we had with the evaluation part of the code base.

For an embedding dim of 2048 it takes roughly 14 hours to complete a single run using 8 H200 GPUs. This was not optimized and is slowed down by heavy logging of a wide range of alignment related metrics. Training was performed in bfloat16 with mixed precision.

#### F.3.1 DETAILS FOR FIGURE 1

This experiment follows our base LLaMA training F.3 and learning rate transfer setup F.2. However these are more recent runs with working validation losses. For the weight decay runs, the learning rate for the input layer and gains is frozen at $\eta = 1.6 \times 10^{-2}$ based on an initial sweep of a global learning rate at width 128 ($1\times$ scale). For the no weight decay runs, this secondary learning rate is set to $\eta = 6.4 \times 10^{-2}$. The best loss values can be seen in Table 1.

**Table 1: Minimum loss values for Figure 1**. Each cell reports the minimum validation loss with the corresponding learning rate in parentheses.

| Model Width | Independent WD Scaling | Standard WD Scaling | No WD |
|---|---|---|---|
| **128** | $4.030\ (8\times10^{-3})$ | $4.032\ (8\times10^{-3})$ | $4.037\ (1.6\times10^{-2})$ |
| **256** | $3.607\ (8\times10^{-3})$ | $3.597\ (1.6\times10^{-2})$ | $3.634\ (3.2\times10^{-2})$ |
| **512** | $3.263\ (1.6\times10^{-2})$ | $3.258\ (3.2\times10^{-2})$ | $3.283\ (6.4\times10^{-2})$ |
| **1024** | $2.991\ (8\times10^{-3})$ | $2.989\ (3.2\times10^{-2})$ | $3.012\ (6.4\times10^{-2})$ |
| **2048** | $2.801\ (8\times10^{-3})$ | $2.798\ (6.4\times10^{-2})$ | $2.815\ (6.4\times10^{-2})$ |

---

[3]https://huggingface.co/meta-llama/Meta-Llama-3-8B

### F.3.2 DETAILS FOR FIGURE 3

This experiment shows `layers.13.feed_forward.w1` for a base learning rate $\eta = 8 \times 10^{-3}$, the optimal value for the $1\times$ width in Figure 1. The learning rate for gains and the input layer is $\eta = 6.4 \times 10^{-2}$. The metrics are logged and plotted every 10 steps, the final 0.5% of training is excluded for numerical reasons (the changes go to zero with the learning rate). The representation based metrics are computed for 1/64th of a batch (corresponding to one micro-batch on the master rank). Other details follow the base setup described in F.3. Additional layers can be seen in Figure 22 and Figure 23 compares the transfer of the absolute and relative representation changes.

### F.3.3 DETAILS FOR FIGURE 4

This experiment shows `layers.13.feed_forward.w1` for a learning rate $\eta = 4 \times 10^{-3}$ and weight decay $\lambda = 0.1$. Note that this is the LR-WD product that gives the best results for standard scaling of the $16\times$ configuration in Figure 1, after applying the $\mu$P scaling to the LR. No $\mu$P scaling is performed between the two configurations, the hyperparameters are exactly the same and the specified learning rate is applied to all parameters. The metrics are logged and plotted every 10 steps, the final 0.5% of training is excluded for numerical reasons. The representation based metrics are computed for 1/64th of a batch (corresponding to one micro-batch on the master rank). See Figure 21 for the alignment ratio of other layers. Other details follow the base setup described in F.3. Similar alignment measurements can be seen in Figure 24 for additional learning rates under independent weight decay scaling.

### F.3.4 DETAILS FOR FIGURE 5

This experiment shows `layers.13.feed_forward.w1` for a learning rate $\eta = 4 \times 10^{-3}$ (after $\mu$P scaling) and weight decay $\lambda = 0.1$. This corresponds to the best configuration for standard weight decay scaling in Figure 1 and the independent scaling run with the same LR-WD product. Metrics are logged and displayed every 10 steps, the final 0.5% are excluded for numerical reasons, representation changes are computed on 1/64th of a batch. Other details follow the base setup described in F.3.

### F.3.5 DETAILS FOR FIGURE 6

This experiment follows our base setup for LLaMA training F.3 and learning rate transfer experiments F.2. The learning rate for the gains and input layer is fixed at $\eta = 1.6 \times 10^{-2}$. The modified learning rate schedules are applied to all parameters, not only the width dependent ones. The scaling factor for `exp50` and `decayaway` varies between widths as in $\mu$P, they have no effect at the $1\times$ scale. The additional warmup factors are applied on top of the existing 10% warmup and linear decay. For the longer 50% linear warmup, the run corresponding to $\eta = 4 \times 10^{-3}$ was repeated due to sudden divergence half way through, which did not occur in the repeated run with exactly the same seed and configuration. The loss values can be seen in Table 2 and are roughly comparable between methods. The exponential schedules perform marginally worse here but are not extensively tuned.

**Table 2: Minimum loss values for Figure 6.** Each cell reports the minimum loss (average of last 100 train steps) for each method and the corresponding learning rate in parenthesis.

| | Model Width | | |
| --- | --- | --- | --- |
| **Method** | **$1\times$ (128)** | **$4\times$ (512)** | **$16\times$ (2048)** |
| Independent $\mu$P scaling + 10% | 3.947 ($8\times10^{-3}$) | 3.232 ($8\times10^{-3}$) | 2.792 ($8\times10^{-3}$) |
| No $\mu$P + 10% | 3.955 ($8\times10^{-3}$) | 3.237 ($4\times10^{-3}$) | 2.785 ($4\times10^{-3}$) |
| No $\mu$P + 50% | 3.955 ($1.6\times10^{-2}$) | 3.252 ($4\times10^{-3}$) | 2.808 ($4\times10^{-3}$) |
| No $\mu$P + 10% + Decayaway | 3.953 ($8\times10^{-3}$) | 3.239 ($8\times10^{-3}$) | 2.800 ($8\times10^{-3}$) |
| No $\mu$P + 10% + Exp50 | 3.953 ($8\times10^{-3}$) | 3.245 ($8\times10^{-3}$) | 2.808 ($8\times10^{-3}$) |

### F.3.6 DETAILS FOR FIGURE 19

These experiments use the LionAR optimizer shown in Algorithm 2. The hyperparameters are $\beta = 0.9$, $\nu = 0$ (no Nesterov momentum). We follow the setup for learning rate transfer experiments from Section F.2, i.e., we sweep a global learning rate at the narrowest width and then fix this while only changing the learning rate of the width-varying layers during the sweeps shown. The global learning rate was swept with the AdamW compatibility mapping of the learning rate from Kosson et al. (2024a). This resulted in $\eta = 0.013$ and $\gamma = 0.016$ for 100 steps, $\eta = 0.018$ and $\gamma = 0.032$ for 1,000 steps and $\eta = 0.013$ and $\gamma = 0.032$ for 20,000 steps. The learning rate schedule is linear decay to zero with 10% linear warmup for the 1,000 and 20,000 step experiments. For the 100 step experiments no learning rate schedule is applied. The losses for the 100 and 1,000 step runs are averaged over 5 runs with different seeds.

It is unclear why the widest network does not perform best for the 100 step training. We note that the validation loss is shown, so perhaps the wider networks do not generalize as well from the relatively few training examples it sees. It could also be that the learning rates for the embedding and gains (which are tuned for the narrowest width) do not transfer well here.

### F.4 RESNET EXPERIMENT CONFIGURATION

Our ResNet He et al. (2016) experiments are performed using a modified version of PyTorch Image Models (Wightman, 2019). All networks are 50 layers deep with a bottleneck block configuration, specifically the original post-norm variant but with the downsampling on the residual stream performed on the 3x3 convolutions ("ResNet v1.5"). When scaling the width we scale all hidden dimensions of all layers evenly relative to the original network, going down to $0.25\times$ width up to $4\times$. The $4\times$ variant is fairly large for a convolutional network with roughly 384M parameters.

The training task is image classification using cross-entropy loss. We train on ImageNet-1k (Russakovsky et al., 2015) for 100 epochs. Data augmentation consists of random cropping and random horizontal flips.

Optimization is performed using the PyTorch variant of AdamW. The learning rate varies between experiments. Other hyperparameters are $\beta_1 = 0.9$, $\beta_2 = 0.999$, $\epsilon = 10^{-8}$, $\lambda = 0.1$ (before independent weight decay scaling). Unless otherwise stated we use a 1 epoch linear warmup from 0 followed by a cosine decay to 0. The batch size is 1024 spread out over 8 GPUs without the use of synchronized batch normalization during training.

The training of the $4\times$ variant takes roughly 10 hours to complete using 8 H200 GPUs. This was not optimized and is slowed down by heavy logging of a wide range of alignment related metrics. Training was performed in bfloat16 with mixed precision.

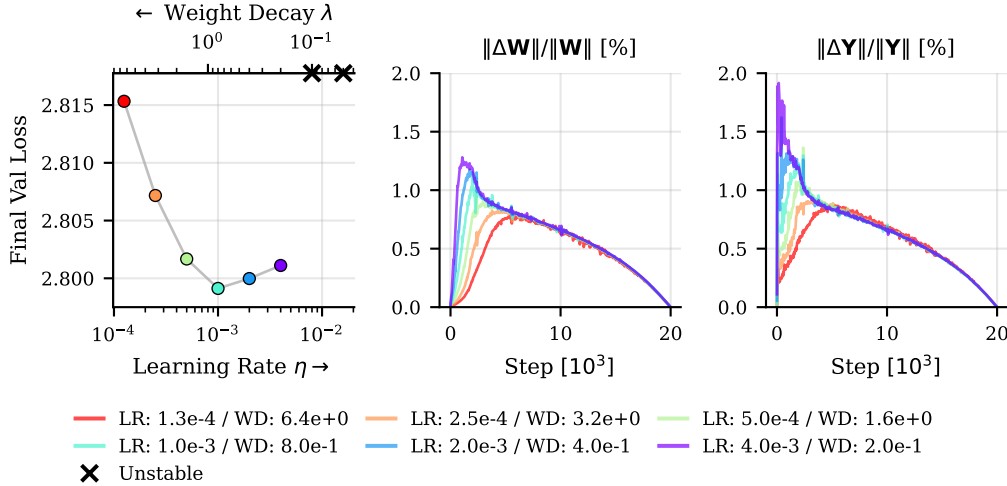

**Figure 17: The scheduling effects of higher weight decay and lower learning rates.** LLaMA experiments where the weight decay and learning rate are varied while their product is kept constant. The effects on the final loss are relatively small (left panel) and likely correspond to using different learning rate schedules which could alter the relative weight updates (middle) and representation changes (right) in a similar manner. For example lower weight decay values have an effect similar to using stronger or longer warmup schedules. Updates shown for a single layer `layers.13.feed_forward.w1`.

## G   WEIGHT DECAY VS LEARNING RATE

Our framework in Equation 4 is expressed in terms of relative weight updates $\|\Delta W\|/\|W\|$. Although this can be applied directly by controlling the relative updates explicitly as done in Appendix H, our main experimental setups with AdamW do not do this. Instead the initialization norm, learning rate and weight decay jointly modulate the size of the relative weight update over time in a complex manner (see Appendix D). The steady-state size of the relative weight updates is determined by the product $\eta\lambda$ of the learning rate $\eta$ and weight decay $\lambda$ as described by Equation 3 (this requires scale invariance or similar properties to hold exactly). This steady-state behavior dominates long training but it is still interesting to understand the secondary effects of varying the learning rate and weight decay while keeping their product fixed. We note that this was explored before in Kosson et al. (2024b) and our findings largely mirror theirs with a couple of new observations.

**Experimental setup:**   We sweep different learning rate and weight decay pairs $(\eta, \lambda)$ in our standard LLaMA AdamW training setup for width 2048 (see Appendix F.3). The product $\eta\lambda$ is kept constant at the value that corresponds to the best configuration for independent weight decay scaling in Figure 1, $\eta\lambda = 8 \cdot 10^{-4}$. The learning rate and weight decay values reported here are the actual values used in the optimizer, after any $\mu$P scaling or similar transformations. We only change the values for the width-dependent weight matrices, the hyperparameters for other parameters like the embedding input layer and gains are kept constant.

**Effect on the loss:**   The left panel of Figure 17 shows how the loss varies with the learning rate and weight decay when their product is fixed. We can see that the loss does vary slightly, but the changes are relatively small compared to changing the product $\eta\lambda$ significantly as happens in Figure 1. The independent weight decay scaling in Figure 1 results in $(\eta, \lambda) = (5 \cdot 10^{-4}, 1.6)$ which is slightly suboptimal here. In contrast the best value for standard weight decay scaling is $(\eta, \lambda) = (6.4 \cdot 10^{-2}, 0.1)$ which has a lower $\eta\lambda$ product and is thus not shown here, but note the significantly lower weight decay value and what this implies in light of the following observations.

**Scheduling Effects:**   We believe the biggest impact of varying the weight decay and learning rate when their product is fixed are scheduling effects. These work similar to modifying the learning rate schedule which can be seen in the middle and right panels of Figure 17. Note in particular how higher weight decay values have a warmup-like effect on both the relative weight updates and relative

representation changes as was also discussed in Section 6. Despite including a standard 10% linear warmup, the violet run with $(\eta, \lambda) = (4 \cdot 10^{-4}, 0.2)$ has large updates at the start of training. Further increasing the learning rate results in even larger updates which are likely the cause of the unstable training we observe for those values. Kosson et al. (2024a) hypothesized that the primary benefit of learning rate warmup is to prevent these large initial representation changes.

We note that the scheduling effects are strongly influenced by the initialization norm. Increasing the initialization norm decreases the size of the initial relative updates in a similar manner as lowering the learning rate and increasing the weight decay. For a perfectly scale-invariant weight, for example when a normalization layer immediately follows a layer, the scheduling effects of varying the learning rate and the initialization norm become exactly the same (up to numerical limitations, optimizer considerations like the $\epsilon$ in Adam, and so on).

**Effects on the weight and representation norms:** Using higher learning rates and lower weight decay values generally results in higher weight norms. For a scale-invariant weight, Kosson et al. (2024b) predicts $\|W\|_{\text{RMS}} \rightarrow \sqrt{\eta/\lambda}$ for sufficiently long training with fixed hyperparameters. The representation norms are directly scaled by the weight norm (for a given direction or weight alignment), but also affected by other operations such as normalization layers and non-linearities. When learnable gains are used, the network has some freedom to control the representation norms independently of the weight norms. As a result weight decay does not necessarily directly constrain the representation norms. This is the case in our LLaMA experiments where there is one learnable gain in the RMSNorm at the start of each residual branch, both for attention and MLP blocks. As is common practice, we do not apply weight decay to the gains allowing them to be learned freely.

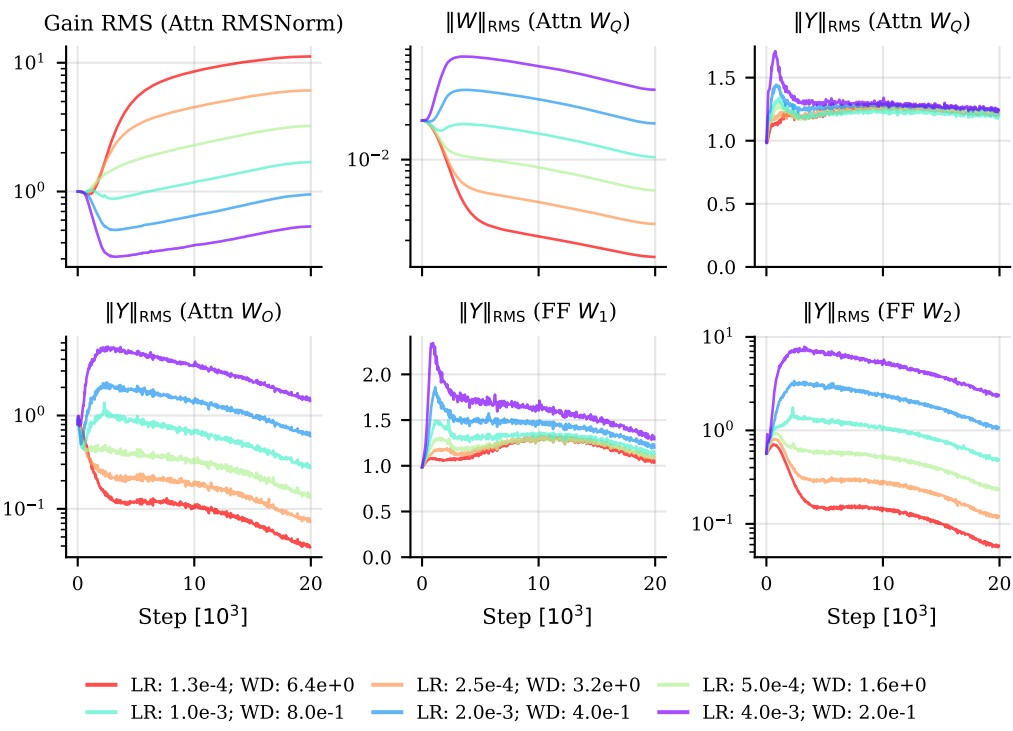

**Figure 18: The behavior of weight and representation norms for different weight decay values.** The top row shows how the network uses the learnable gain (upper left) in the RMSNorm to counteract differences in the weight magnitude of the following layer (upper center) in order to get query representations of a fixed magnitude (upper right). With only one gain per residual branch the network can not further control the output magnitudes of the residual branch (lower left). The MLP block behaves similar, the input magnitude to the SiLU non-linearity is roughly preserved (lower center) but the residual branch output magnitude is not (lower right). Metrics shown for block 13.

Figure 18 shows measurements of different weight and representation norms over time during LLaMA training with different $(\eta, \lambda)$ values. We observe that the network can learn to use gains to counteract changes in the weight norm due to weight decay in a way very similar to the rescaling transformations described in Section 2. This seems to happen with the attention softmax in particular. Here the gain (upper left panel) learns to perfectly counteract changes in the weight norm of the query transform (upper center panel) resulting in a stable query magnitude (upper right panel). Altering the magnitude of the query and key representations would change the selectiveness of the attention in a potentially detrimental way, similar to changing the softmax temperature. With only one gain per residual branch shared between the key, query, and value transformations, the network can not further learn to counteract changes in the weight norm of the attention output transformation. As a result the output magnitude of the residual branch (lower left panel) is strongly affected by weight decay.

We observe a similar pattern in the MLP block though it is unclear if the network is using the gain to keep the input magnitude of the SiLU (bottom center panel) consistent or trying to balance the output magnitudes of the attention (lower left panel) and MLP blocks (lower right panel). Adding an extra gain per residual branch or one per linear layer would allow the network to learn the representation norms directly, but we do not explore the effects or potential benefits of this here.

**Other effects:** If the learning rate hyperparameter for weight matrices is also applied to other parameters like gains and biases, this becomes a strong secondary effect of changing the weight decay as discussed in Kosson et al. (2024b). There could also be effects related to numerical precision. When using floats with wide exponent ranges, representations of different magnitudes can likely be encoded accurately. However, this is not the case for fixed point and narrower float formats which might have significant issues if this is not compensated for.

**Algorithm 2** LionAR (Kosson et al., 2024a): A rotational version of the Lion optimizer (Chen et al., 2023) that replaces weight decay with projections onto a sphere of fixed radius, controlling relative weight updates directly. This version does not include the AdamW learning rate compatibility mapping from Kosson et al. (2024a), making the relative updates directly proportional to the learning rate. Note that this makes the (relative) learning rate more sensitive than in AdamW (it should be scaled more like the square root of the AdamW learning rate with batch size or network width).

---

**Require:** Relative learning rate $\eta_t$, scalar learning rate $\gamma_t$, momentum $\beta$, Nesterov factor $\nu$
1: **Initialize:** Time step $t \leftarrow 0$, parameter vector $\boldsymbol{\theta}_0 = [\boldsymbol{\theta}^{(1)}, \ldots, \boldsymbol{\theta}^{(P)}]$, divided into sub-vectors at the granularity of individual matrix rows, convolutional filters, embeddings, or biases/gains, momentum vector $\boldsymbol{m}_0 \leftarrow 0$
2: **while** stopping criteria not met **:**
3:     $t \leftarrow t + 1$
4:     $[\boldsymbol{g}_t^{(1)}, \ldots, \boldsymbol{g}_t^{(P)}] \leftarrow$ Mini-batch gradient w.r.t. $\boldsymbol{\theta}_{t-1}$, divided into sub-vectors like $\boldsymbol{\theta}$
5:     **for** $p \in \{1, \ldots, P\}$ **:**
6:         $\boldsymbol{m}_t^{(p)} \leftarrow \beta \boldsymbol{m}_{t-1}^{(p)} + (1 - \beta) \boldsymbol{g}_t^{(p)}$
7:         $\boldsymbol{u}_t^{(p)} \leftarrow (1 - \nu) \boldsymbol{m}_t^{(p)} + \nu \boldsymbol{g}_t^{(p)}$
8:         **if** $\boldsymbol{\theta}^{(p)} \in \mathbb{R}^C$ is a weight matrix row, embedding, or convolutional filter **:**
9:             $\hat{\boldsymbol{\theta}}_t^{(p)} \leftarrow \boldsymbol{\theta}_{t-1}^{(p)} - \eta_t \cdot \|\boldsymbol{\theta}_0^{(p)}\| \cdot \text{sign}(\boldsymbol{u}_t^{(p)})/\sqrt{C}$      *# Scale update to be prop to sphere radius*
10:             $\boldsymbol{\theta}_t^{(p)} \leftarrow \hat{\boldsymbol{\theta}}_t^{(p)} \cdot \|\boldsymbol{\theta}_0^{(p)}\|/\|\hat{\boldsymbol{\theta}}_t^{(p)}\|$           *# Project value back onto the sphere*
11:         **else:**
12:             $\boldsymbol{\theta}_t^{(p)} \leftarrow \boldsymbol{\theta}_{t-1}^{(p)} - \gamma_t \cdot \text{sign}(\boldsymbol{u}_t^{(p)})$   *# Standard update without weight decay for gains and biases*

---

## H   Training with Constrained Weight Norms

As an alternative to weight decay, we can instead constrain the weight norms to their initialization values using projections after each step. Although this approach is still not very common in practice, it has shown promise for training diffusion models (Karras et al., 2024), in reinforcement learning (Lyle et al., 2024) and transformer language model training (Loshchilov et al., 2025). By doing this we can eliminate the more complex scheduling effects of weight decay and have the learning rate directly determine the relative weight update $\|\Delta \boldsymbol{W}\|/\|\boldsymbol{W}\|$.

We use a variant of the Lion optimizer (Chen et al., 2023) similar to LionAR (Kosson et al., 2024a) but without the additional learning rate adjustments for AdamW compatibility, see Algorithm 2. In this setup the learning rate directly controls the size of the relative weight updates throughout training, there is no convergence to rotational equilibrium or similar. The weight magnitudes remain fixed throughout training, keeping the norms close to what the $\mu$P initialization strategy prescribes rather than letting them drift away as in standard training.

Figure 19 shows how the optimal learning rate stays roughly constant across width for longer runs. This directly contradicts $\mu$P's prediction for the scaling of the relative weight update size, see Equation 6 (which we emphasize is consistent with multiple prior $\mu$P works). This behavior can be explained by the changes in the alignment ratio as our framework predicts. Figure 20 compares how the relative representation change is preserved when $\mu$P scaling is applied vs no $\mu$P scaling. We can see that due to the alignment ratio being roughly width independent for most of training, not applying any scaling better preserves the relative representation change. The exception is early on when the representation change can spike without $\mu$P, but here this is not detrimental enough to shift the optimal learning rate. Since no weight decay is applied we can not get an additional warmup effect from independent weight decay scaling, but exponential warmup strategies like those discussed in Section 7 could be applied if this becomes an issue for wider networks.

The LionAR experiments allow us to isolate the effects of the relative weight update without the other effects of weight decay (Appendix G). The findings are consistent with our interpretation of independent weight decay scaling, it helps by keeping the relative weight update later in training similar across network sizes in contrast to what $\mu$P prescribes. The experiments also highlight the wide applicability of our framework by showing it holds for a different optimizer (Lion) and a different form of explicit weight constraints (on top of the weight decay and no weight decay settings discussed before). Finally training with constrained weight norms might be a setting where our warmup interpretation of $\mu$P with independent weight decay has a direct practical application.

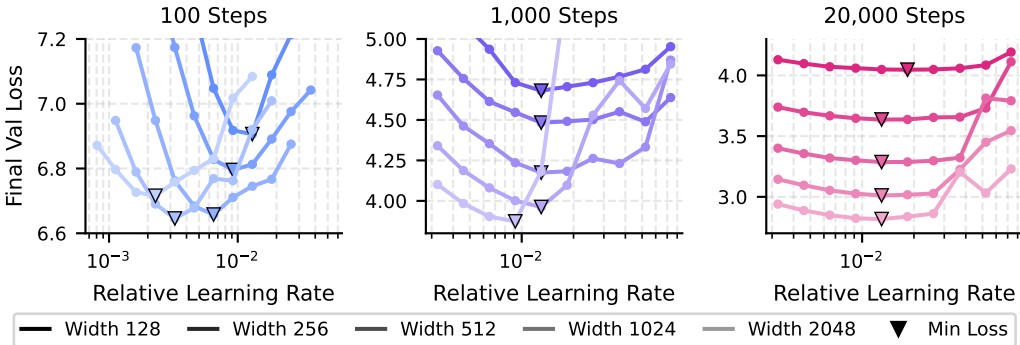

**Figure 19: The relative weight update size should be roughly constant across width for longer runs rather than scaled as $\mu$P prescribes**. LLaMA experiments with the LionAR optimizer (Algorithm 2) where no $\mu$P-like scaling is applied to the relative learning rate across widths. We note that applying $\mu$P's prescribed scaling would be mathematically equivalent to shifting the curves to the right by $(\text{width}/128)^{\frac{1}{2}}$. The weights are constrained to their initialization norms and no weight decay is applied. Note how there is only minimal shift in the optimal learning rate across widths for the 1,000 and 20,000 step runs. In contrast the optimal learning rate shifts significantly for the 100 step run, similar to $\mu$P's prediction from Equation 6. The prediction is that the relative weight update, and thus the relative learning rate, should decrease with the square root of the network width. See F.3.6 for experimental details.

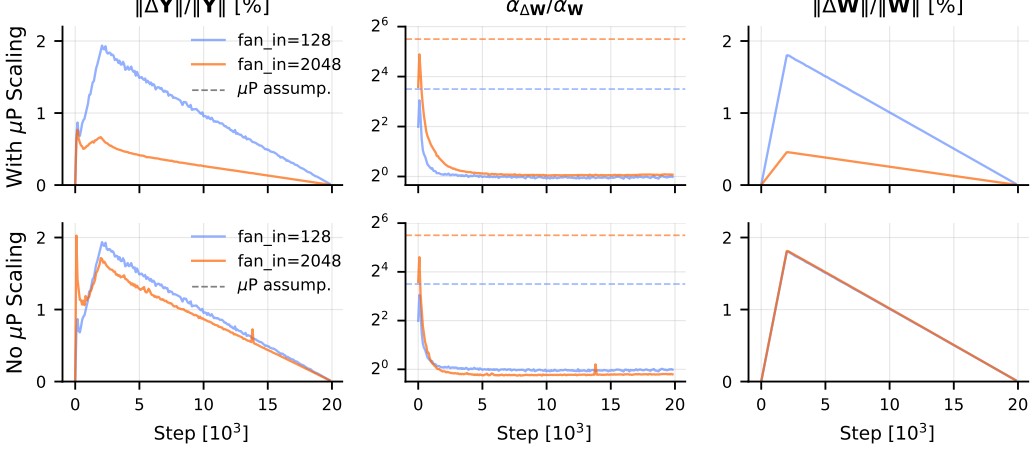

**Figure 20: Not scaling relative weight updates as $\mu$P prescribes better preserves relative representation changes for most of training.** LLaMA experiments corresponding to those shown in the right side of Figure 19. Recall how the relative representation change $\|\Delta \boldsymbol{Y}\|/\|\boldsymbol{Y}\|$ is the product of the alignment ratio $\alpha_{\Delta \boldsymbol{W}}/\alpha_{\boldsymbol{W}}$ and relative weight update $\|\Delta \boldsymbol{W}\|/\|\boldsymbol{W}\|$ (see Equation 4). In the upper row we compare the metrics for a relative learning rate of 0.018 for width 128 and 0.0046 for width 2048, matching $\mu$P's prescribed scaling of $1/\sqrt{m} = 1/4$ from Equation 6. In the lower row we do not apply any scaling using 0.018 for both widths. Note how the latter approach results in a better transfer of the relative representation change $\|\Delta \boldsymbol{Y}\|/\|\boldsymbol{Y}\|$ aside from a very brief period at the start of training. Metrics shown for a single layer `layers.13.feed_forward.w1`.

# I  SUPPLEMENTARY FIGURES

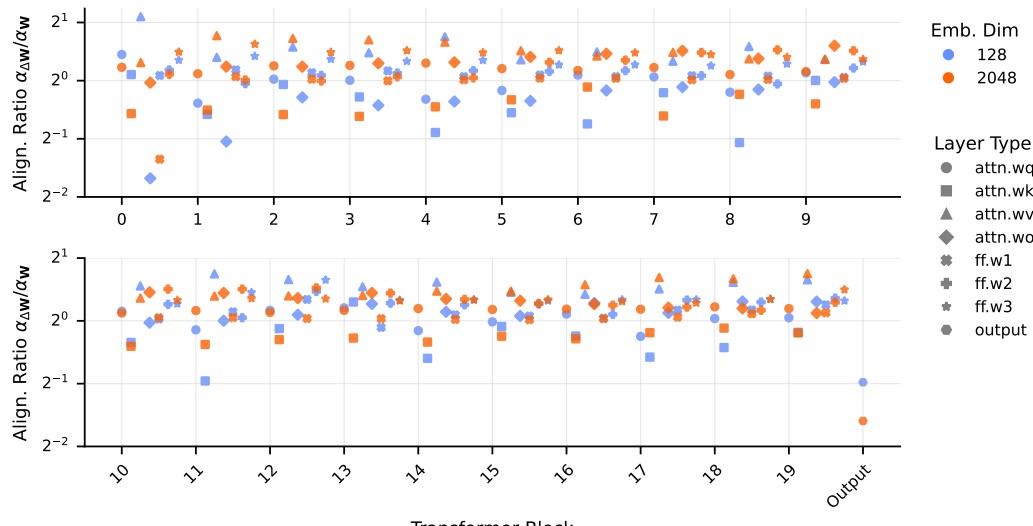

**Figure 21: Halfway through training the alignment ratio of all layers is approximately one**. The alignment ratio for each layer in the LLaMA experiments shown in Figure 4 measured at step 10,000. The ratio is similar across widths, contrary to $\mu$P's assumptions. The values should be contrasted to $\mu$P's assumed alignment ratio of $\sqrt{C}$ where $C = d_{\mathrm{emb}}$ for all layers except ff.w2 where $C = 3d_{\mathrm{emb}}$. This would give values of roughly $\sqrt{2048} = 2^{5.5} \approx 45$ for most orange markers. The gap between the blue and orange markers could also be compared to $\mu$P's predicted value of $\sqrt{2048/128} = 4$, where the orange marker should be higher.

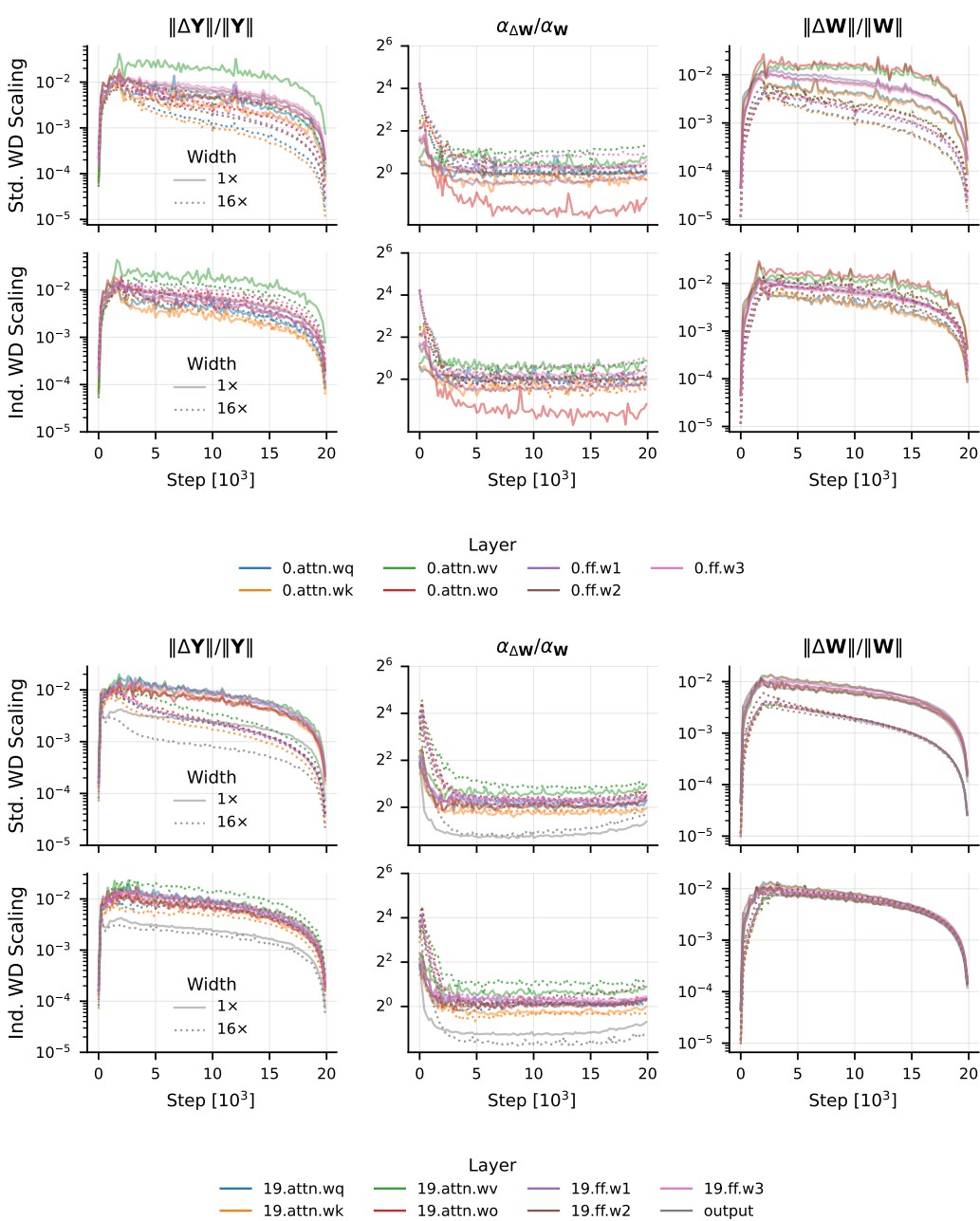

**Figure 22: Comparing the transfer of relative representation changes across layers.** An extension of Figure 3 showing the behavior for additional layers in LLaMA, specifically those in the first transformer block (upper half of figure) and last transformer block (lower half of figure). The relative representation change (RRC) transfers well under independent weight decay scaling for almost every layer, unlike standard scaling. The worst RRC transfer for independent scaling occurs in `0.attn.wo` where the alignment ratio for the narrower network stabilizes at a value notably lower than for the wide network for unknown reasons. Compared to Figure 3, each line is further downsampled to 100 roughly equally spaced points without averaging.

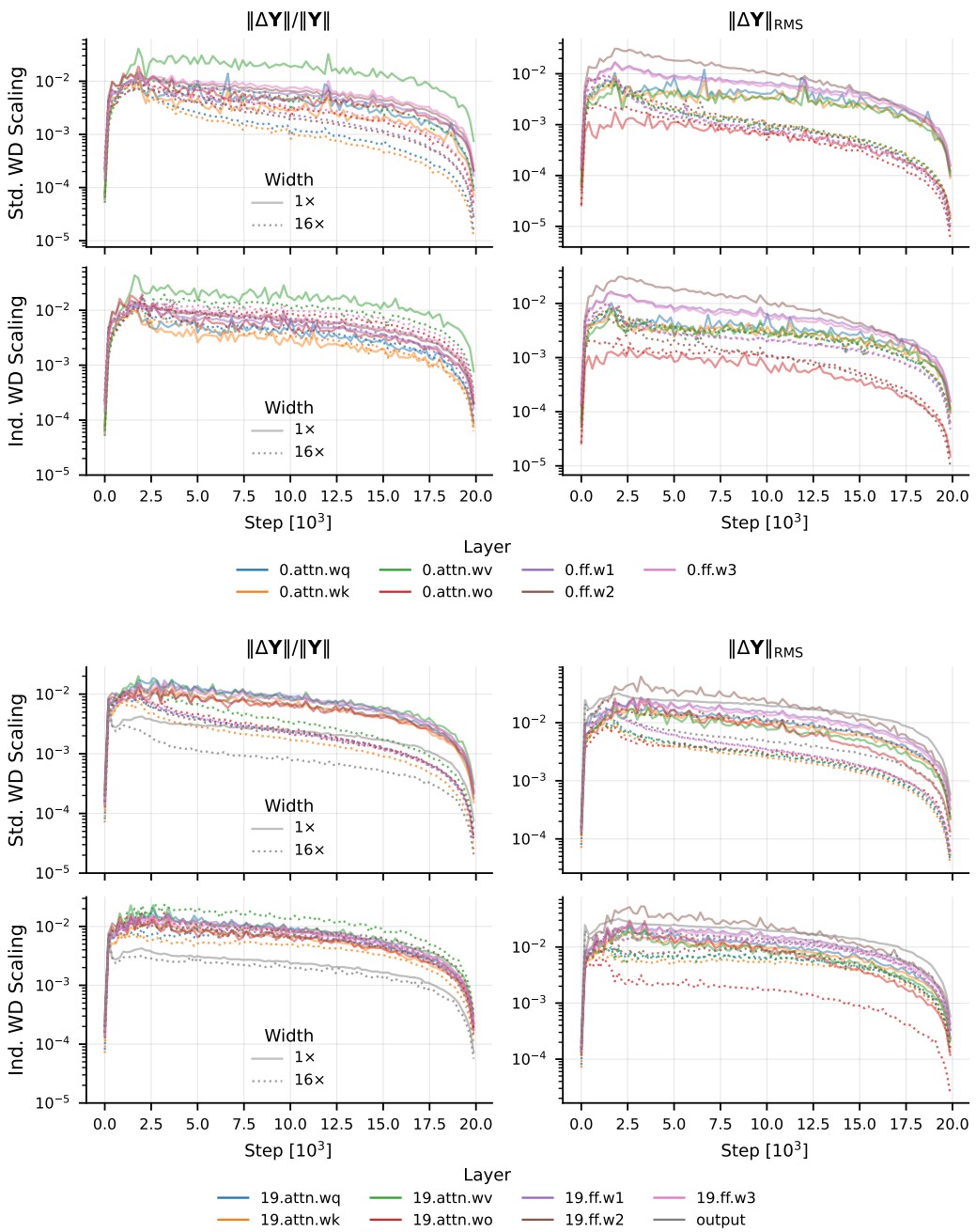

**Figure 23: Comparison of relative and absolute representation changes.** An extension of Figure 3 comparing the transfer of the relative vs absolute representation change across multiple layers in LLaMA, specifically those in the first transformer block (upper half of figure) and last block (lower half of figure). The relative representation change transfers $\|\Delta Y\|/\|Y\|$ well for independent weight decay scaling as shown before in Figure 22. Neither metric transfers for standard weight decay scaling. The absolute representation change $\|\Delta Y\|_{\mathrm{RMS}}$ does not normalize for the magnitude of the inputs to the layer $\|X\|$ or the size of the weights $\|W\|$. For this reason it does not isolate the contribution of a single layer and the transfer characteristics can vary across layers depending on where they are located relative to normalization layers and gains as argued in Section 2.3. Note the poor transfer for `0.ff.w2`, `19.attn.w0` and `19.ff.w2` in particular, even for independent scaling where the LR transfers well (these are residual branch output layers). Compared to Figure 3, each line is further downsampled to 100 roughly equally spaced points without averaging.

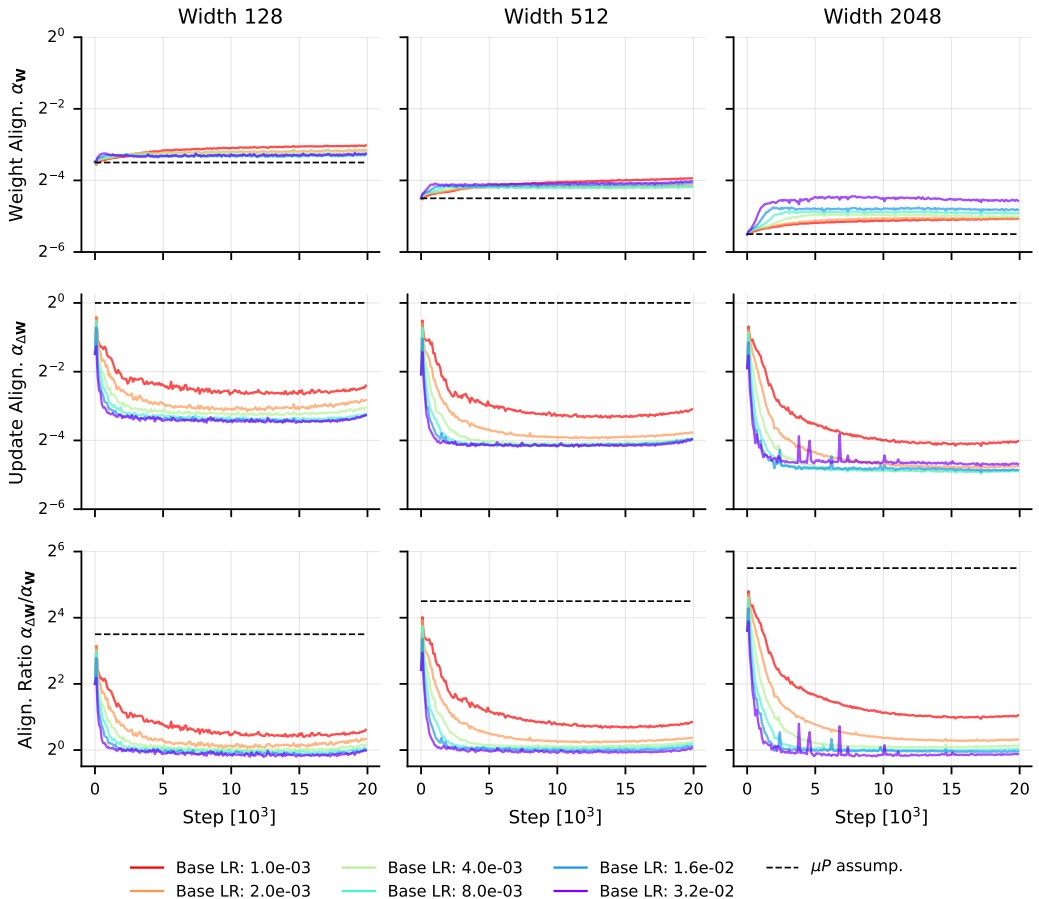

**Figure 24: Alignment behavior across widths and learning rates**. Measurements of the weight alignment, update alignment and alignment ratio for LLaMA training for different model widths and learning rates. Independent weight decay scaling is used, the runs correspond to those shown in the left panel of Figure 1 where the optimal base learning rate is $\eta_{\text{base}} \approx 8 \cdot 10^{-3}$. Note how higher base learning rates result in faster changes in the alignment ratio. This is expected as meaningful updates to the network are needed to change the behavior (for learning rate zero the alignment would never change). Other details follow those for Figure 4 in Appendix F.3.3, except the lines are further downsampled to 200 roughly equally spaced points without averaging.

