# OpenReview forum: "Weight Decay may matter more than µP for Learning Rate Transfer in Practice"
_ICLR.cc/2026/Conference — ICLR 2026 Poster_

### Official Review · Reviewer_aDn7 · 2025-10-30

**Soundness:** 4
**Presentation:** 3
**Contribution:** 2
**Rating:** 4
**Confidence:** 3

**Summary:**

The paper discusses a previously known fact that $\mu$P, which is commonly thought to give learning rate transfer between different model widths, actually does not unless paired with independent weight decay.
Authors highlight that $\mu$P's alignment assumptions, when viewed in terms of relative updates, are incompatible with IndepWD. This can be reconciled by the fact that ultimately $\mu$P's alignment assumptions are not correct.
Authors show empirically that IndepWD enables transfer of 'relative representational changes', whereas $\mu$P does not.
Finally authors propose particular learning rate warmup schedules as alternatives to $\mu$P, and show that these are comparable to $\mu$P.

**Strengths:**

To my knowledge, emphasising the role of _relative_  representational changes in learning-rate transfer is novel, as is the perspective that weight decay, rather than the learning rate itself, governs successful transfer. The observation that $\mu$P effectively acts as a learning-rate warm-up schedule is particularly insightful, as is the clarification that $\mu$P conflicts with the usual recommendation to keep $\eta\lambda = \text{const}$.

The paper convincingly demonstrates how $\mu$P’s infinite-width assumptions fail in practice—since realistic networks do not maintain a small batch-size-to-width ratio—and consequently why $\mu$P alignment can break down.

The authors illustrate their results in both transformers and resnets.

I think the observations in this paper are of great interest to those training neural networks.

**Weaknesses:**

It's not immediately obvious what this paper's contributions are, versus what already existed. I would appreciate the standard bulleted list at the end of the introduction to declare contributions unambiguously. I feel this would improve the clarity, and highlight the novelty of the paper.

I would like to see more justification that RRC are what we should be interested in, as opposed to absolute norms of representations. As far as I can tell, there are not any plots for this. It's also unclear if transfer of the RRC appears for all weights, or just the one shown (from layer 13), possibly a figure to illustrate many/all layers should be included.

I would like to see some tables with performance metrics. Some Tables in the Appendix confirming that optimal performance is similar across different settings ($\mu$P+IndepWD, vs. $\mu$P) would help, as well as performance figures to compare $\mu$P to the proposed learning rate schedules.

**Questions:**

- How did you pick which weight to show for Figures 3, and 4? The alignment ratio is almost exactly 1 for this weight, but it doesn't seem to hold in general based on Figure 16/Appendix G, and thus I'm not sure how valid the approximation for $\alpha_{\Delta W}/\alpha_W$ is (in section 5.2).
- Have you thought about $\mu$P/IndepWD/RRC during fine-tuning? In particular with LoRA, and how your observations interact with work like [1]
- I would be interested to see how the RRC plots behave for different optimizers?

[1]: Hayou, Soufiane, Nikhil Ghosh, and Bin Yu. "Lora+: Efficient low rank adaptation of large models." _arXiv preprint arXiv:2402.12354_ (2024).

Minor points:

- Is there a typo in line 105/106? 'high update alignment means that $w_k$ and $x_b$ tend to point in similar directions'
- What is $\eta_\tau$ in Eq. (13)?
- 'scaling effect' should be defined in Figure 5's caption, or at the very least, the right panel should be explicitly described.


I would generally be in favour of acceptance if the weaknesses and my questions above can be addressed.

---

> ### Author Response · Authors · 2025-11-22
>
> We thank Reviewer aDn7 for their constructive feedback and for providing actionable suggestions to improve the manuscript. We are encouraged that you found our perspective on Relative Representational Changes (RRC) novel and our observations regarding µP insightful. We have incorporated your suggestions into the updated manuscript, including the explicit contribution list and additional performance metrics. Our detailed responses follow below.
>
> ---
>
> ### Weakness 1: Contribution Bullets
> We have added a concise list of contributions to the introduction in the updated manuscript.
>
> ### Weakness 2: Justify RRC over absolute changes, show additional layers
> * We have added a new Figure 21 (Appendix I, page 35) that shows the RRC transfer for more layers (all layers in the first and last transformer blocks). There is slight variation between layers but overall the behavior is quite similar to the example layer we showed before.
> * We have added a new Figure 22 (Appendix I, page 36) that compares the relative representation change (RRC) to the absolute representation change (ARC). The RRC shows a more consistent transfer pattern between layers compared to the ARC.
> * The conceptual reason we use the RRC is that it normalizes and isolates the contribution from a single layer; it does not matter whether the inputs to the layer are large or small (which affects the ARC). We justify this intuitively based on the structure of neural networks in Section 2.3. We have also added explicit measurements of how the network can learn to use gains to counteract changes in the weight norm in the newly added Appendix G, see Figure 17 on page 30 in particular. These measurements provide a strong argument for the RRC since the ARC transfer will depend on details such as whether the gains are placed before or after a particular layer and similar architectural considerations.
>
> ### Weakness 3: Tables with performance numbers
> We have added these to Appendix F.3 (Tables 1 and 2).
>
> The results show that replacing µP with our proposed heuristic warmup schedules achieves comparable learning rate transfer and stability, albeit with a slight difference in final loss (e.g., 2.792 vs. either 2.800 or 2.808 for the widest model). We emphasize that our proposed "Decay-away" and "Exp50" schedules are simplified heuristics intended to test our hypothesis, rather than perfectly tuned replacements. The fact that these simple heuristics recover the vast majority of the performance (and the transfer capability) validates our core claim: µP’s benefit arises from an implicit warmup effect rather than a sustained scaling of the learning rate. Just as there is no theoretical guarantee that the specific implicit schedule induced by µP+IndependentWD is globally optimal, our heuristic approximations are not optimized to the limit, yet they successfully replicate the transfer mechanics.

---

> ### Author Response · Authors · 2025-11-22
>
> ### Questions
> **How do we pick the weight for Figure 3 and 4?** \
> We picked this layer early on as a representative sample and used it throughout the paper for consistency. We do log these metrics for every layer and looked through them to ensure it was fairly representative. We have added plots for additional layers as you suggested in the weakness section and believe they show similar trends.
>
> Regarding the approximation for the alignment ratio, we note that most layers lie between $2^{-0.5} \approx 0.7$ and $2^{0.5} \approx 1.4$. While this is not exactly 1, it stands in stark contrast to the µP assumption, which says the ratio should scale with width (approx. $\sqrt{2048} \approx 45$ for the wide network). The RRC transfer is also not dependent on the alignment ratio being exactly 1 but rather on it being similar for the narrow and wide networks. We have also found that the value of the alignment ratio does depend slightly on the learning rate; for values significantly below the optimal learning rate the values tend to be a bit higher (see the newly added Figure 23, page 37).
>
> **Applicability to fine tuning?**\
> Thank you for this reference, we have not looked into this but believe this is a promising direction that could be interesting to explore in the future. How and whether to apply weight decay to fine tuning seems like an interesting problem in general with additional challenges. For example we expect the weight norm to significantly matter here compared to pre-training, since it needs to have the correct size relative to the pretrained weights.
>
> **RRC for different optimizers?**\
> We have added measurements for training with Lion with fixed weight norms in the new Appendix H (see Figure 19, page 33). The behavior is similar in terms of the alignment ratio. A key advantage of this setup is that there is no weight decay and the learning rate directly controls the relative weight update. Thus there is no need for the relative weight update to converge or similar, simplifying the overall dynamics.
>
> ### Minor Points
> We appreciate the attention to detail and have corrected these in the text:
> * Line 105/106: Fixed.
> * Eq 13: $\eta_t$ denotes the learning rate at time $t$ (including the schedule).
> * Figure 5: Caption updated to define the scaling effect.

---

### Official Review · Reviewer_x5KA · 2025-10-31

**Soundness:** 4
**Presentation:** 4
**Contribution:** 3
**Rating:** 8
**Confidence:** 2

**Summary:**

The paper studies Maximal Update Parameterization ($\mu$P), a learning rate scaling for transfer between different model widths, and its relation to weight decay. To do this, the authors develop a framework based on relative updates (i.e., the size of updates proportional to the weights). They use this to show that independent weight decay effectively counteracts $\mu$P scaling, which counterintuitively leads to an improvement in transfer. They show that this is because the alignment assumptions of $\mu$P do not hold as training progresses in practice. However, because of this, independent weight decay stabilizes feature learning by counteracting $\mu$P and maintaining the same relative representation changes across network widths. The authors show that early in training, $\mu$P provides a benefit by acting as an effective learning rate warmup and support this by showing that a stronger learning rate warmup can replace $\mu$P’s learning rate scaling with similar results.

**Strengths:**

The paper studies $\mu$P scaling and weight decay, a subject of significant practical relevance for hyperparameter transfer. The authors identify why $\mu$P scaling works with independent weight scaling, diverging from the reasons assumed by the theory used to develop it. The framework, approach, and results appear original. The paper supports its claims with theory and solid experiments using LLaMa and ResNet. The paper is well-written, clear, and well-motivated; the overall presentation of the paper is excellent.

**Weaknesses:**

As someone who is less familiar with this area, I found it somewhat challenging to follow and get an intuition for the framework being introduced—in particular, comparing and contrasting between update alignment, weight alignment, relative weight updates, alignment ratio, and relative representation. The summaries and discussion helped with getting an overall understanding, but it would have been helpful to perhaps use less jargon or to give more intuition for the different terms.

**Questions:**

How is update or weight alignment related to data eigenvector alignment (if it is at all)? I’m wondering if the “silent alignment” effect (Atanasov et al., 2022) is related to the alignment and muP “warmup” discussed here or if these are separate concepts.

---

> ### Author Response · Authors · 2025-11-25
>
> We thank reviewer x5KA for their review. We are encouraged by the positive assessment of our work, highlighting its originality, practical relevance, and presentation quality.
>
> We appreciate the feedback on Section 2. We need to introduce a lot of quantities to bridge the gap between µP theory (which has a reputation for being somewhat inaccessible) and prior work on weight decay. We will review this section to see where we can simplify the language or add intuitive clarifications.
>
> Regarding the silent alignment effect from Atanasov et. al: We looked into this but were not able to formulate a direct connection between the weight and update alignment discussed in our manuscript and the eigenvector alignment described by Atanasov et al.
> * The eigenvector alignment describes the overlap between the NTK and the target labels.
> * The weight and update alignment describe the geometric alignment between the inputs to a layer and the weights or updates for the layer, respectively.
>
> The definitions are not directly related, but there could be a deeper connection, particularly to the update alignment. We hypothesize that the update alignment is affected by the similarity of the gradients between different samples (through the Signal-to-Noise ratio described in Appendix C), something that the NTK also measures. The changes in all three alignment metrics are also driven by learning and could therefore have a common cause in that sense. We will add a brief note in the related work section acknowledging this potential connection.

---

### Official Review · Reviewer_rEs9 · 2025-11-02

**Soundness:** 2
**Presentation:** 3
**Contribution:** 3
**Rating:** 4
**Confidence:** 4

**Summary:**

The paper studies learning-rate transfer under the Maximal Update Parameterization ($\mu$P) and the role of weight decay. $\mu$P scales learning rates to keep feature updates stable across widths, supporting hyperparameter transfer from small to large models. The paper observes that $\mu$P’s theoretical alignment assumptions only hold early in training, especially in large-batch settings common in practice.

Empirically, the authors show that independent weight decay, where the product of learning rate times weight decay $\lambda \eta$ is scale-invariant, is essential for stable learning-rate transfer in large models, while standard weight decay fails to provide transfer.

The authors further demonstrate that the primary effect of $\mu$P in practice is similar to an implicit learning-rate warm-up. They show that explicit warm-up strategies can partly replicate the benefits of $\mu$P.

**Strengths:**

- Understanding why the learning-rate transfer works under $\mu$P, and what happens under $\mu$P scaling over long training horizons, are certainly interesting and of fundamental importance. In this sense, the experiments on the alignment ratio are a valid contribution, showing that no matter how weight decay is scaled (Figure 3, middle plot), the alignment ratio has a very weak dependence on the width.

- The experiments are very well documented in the appendix, where there are details for each Figure.

- The experiments on achieving good transfer when the width scaling is incorporated only in the warm-up phase are intriguing. It seems to suggest that $\mu$P scaling is mainly required close to initialization (although there is increased instability, probably due to the fact that the updates explode with width after the warm-up phase). These warm-up experiments isolating early-training effects suggest that $\mu$P’s impact may be concentrated near initialization.

**Weaknesses:**

A central narrative of the paper is that independent weight decay “contradicts” $\mu$P by overriding its scaling. Quoting from the paper: *“This fundamentally differs from $\mu$P’s prescribed scaling in Equation 6, meaning that independent weight decay scaling contradicts $\mu$P by eventually overriding its update scaling.”*

My interpretation differs: under *standard* weight-decay scaling, the contribution of weight decay vanishes as width grows (as also noted in [1]: *“However, the implied EMA timescale now changes with model size, while we hypothesize that the optimal timescale should not vary with model size.”*), which breaks width-invariance of feature updates — a core $\mu$P premise. From this perspective, independent weight decay does not contradict $\mu$P, but rather restores its intended property that the components of the feature updates remain width-invariant. Under standard weight decay, weight-decay’s contribution vanishes in the large-width limit (this can be shown via Tensor-Programs-style calculations), which, in my view, explains why the optimal learning rate shifts to the right in Figure 1 (middle plot): increasing the learning rate partially compensates the vanishing contribution of weight decay.

That said, the paper raises an important possibility: over long training horizons (e.g., when total steps scale with width), the training trajectory may move sufficiently far from initialization that $\mu$P’s assumptions no longer hold. This is a compelling direction. However, the conceptual leap from “assumptions break late in training” to “learning-rate transfer fails” would benefit from further justification and clearer separation from the effect of vanishing weight decay. In fact, the paper’s experiments seem to show deviations from the $\mu$P regime under both independent and standard weight-decay scaling (as observed in the alignment ratio plots, Figure 3 middle). To me, this suggests instead that the vanishing contribution of weight decay with width is the cause of poor learning-rate transfer. This does not contradict $muP$'s philosophy of maintaining scale-invariant contributions to the feature updates.


**Clarification request**

Is Figure 4 conducted under a setup that actually uses $\mu$P? In the appendix, it is written: *“the hyperparameters are exactly the same and the specified learning rate is applied to all parameters”*, which seems to suggest that **no $\mu$P scaling** was applied. If so, how can this be used to claim $\mu$P assumptions break, if $\mu$P was not used? I would appreciate a clarification.

**Warm-up discussion**

In light of the above discussion, the result that learning-rate warm-up alone cannot replace $\mu$P (Figure 6 left) is not surprising. This is because the exploding updates with width cause the optimal learning rate to shift left.

**Overall**

This paper contains valuable experiments, but I believe the narrative requires restructuring. In particular, I would focus more on the investigation of $\mu$P over long training horizons and the role of weight decay.


[1] Xi Wang, Laurence Aitchison, How to set AdamW's weight decay as you scale model and dataset size.

**Questions:**

How is $\Delta Y / Y$ computed? At the same time step?

---

> ### Author Response · Authors · 2025-11-22
>
> We thank the reviewer for the detailed engagement. We will focus on the core disagreement regarding Independent Weight Decay (Ind-WD) and µP, as we believe clarifying this resolves the other concerns.
>
> ---
> ## Core Arguments
>
> ### 1. Independent Weight Decay overrides µP's explicit scaling rules
> Prior µP literature derives that the Relative Weight Update (RWU) should scale as $\|\Delta W\|/\|W\| \propto 1/\sqrt{C}$ (where $C$ is the fan-in). This is explicitly stated in:
> * Tensor Programs V: Tuning Large Neural Networks via Zero-Shot Hyperparameter Transfer (arXiv:2203.03466), appendix B.3
> * A Spectral Condition for Feature Learning (arXiv:2310.17813), section 5.4
> * Scaling Exponents Across Parameterizations and Optimizers (arXiv:2407.05872), AdaFactor in Table 1 (full alignment case)
>
> Note that these results refer to optimizer modifications that would ensure that the scaling of the RWU persists throughout training, not only early on. Ind-WD results in a constant RWU in equilibrium (invariant to width). This creates a direct contradiction: µP prescribes $1/\sqrt{C}$ scaling, but Ind-WD eventually ensures $O(1)$. Even if one argues Ind-WD is necessary for the spirit of µP, it contradicts the explicit scaling prescribed in the papers above. If µP simultaneously prescribes $1/\sqrt{C}$ scaling (via update scaling) and constant scaling (via Ind-WD), the framework is internally inconsistent. Our work resolves this discrepancy by identifying when each scaling is appropriate based on changes in the alignment ratio.
>
> ### 2. New Evidence: µP scaling fails even without Weight Decay (Appendix H)
> To prove this is not merely about "restoring vanishing weight decay," we introduce Appendix H. In this setup we keep the initialization norms throughout training, in-line with what µP prescribes, rather than letting them change as in standard training. This setup removes Weight Decay entirely, isolating the update scaling dynamics.
>
> We find that constant update scaling gives the best transfer for longer training, the $1/\sqrt{C}$ scaling prescribed by µP is only appropriate for very short runs (New Figure 18). This perfectly mirrors our Ind-WD findings. Since this phenomenon persists in a WD-free environment, the "vanishing WD" hypothesis cannot be the root cause here. The necessity for constant scaling arises because µP's underlying alignment assumptions quickly break down, necessitating a constant RWU to maintain stable representation changes later in training.
>
> ### 3. Our framework unifies these observations without ad-hoc assumptions
> The "vanishing WD" perspective requires a separate, external assumption: that WD contributions to the feature updates must not vanish with width. In weight space this means that the weight decay updates stay finite even as µP makes the gradient updates infinitesimal as the width increases. A clear consequence of this would be that the weight norms must shrink to zero, which in turn causes the representations for a random input to vanish over time. This behavior deviates from what µP prescribes at initialization and contradicts its goal of keeping representation sizes comparable across widths.
>
> In contrast, our Relative Update Framework relies on a single objective: **ensuring stable Relative Representation Changes across widths.**
> * Early in training (alignment varies with width): This objective necessitates µP's $1/\sqrt{C}$ scaling.
> * Late in training (alignment is constant): This objective necessitates constant scaling (Ind-WD).
>
> This single principle explains both the success of µP's LR scaling and the necessity of Ind-WD, without requiring contradictory assumptions about WD behavior. The shrinking norms are acceptable in our framework as long as they can be countered by normalization layers or learnable gains.
>
> **Connection to EMA Timescale:** Our framework also provides an alternative justification for the "EMA Timescale" intuition proposed by Wang & Aitchison 2405.13698, linking it directly to the rate of feature learning. They hypothesize that the effective timescale of training should be constant across model sizes. Since the timescale is essentially an alternative formulation of relative weight update size, our findings explain why their intuition holds: the timescale (and update size) must be constant later in training precisely because the alignment ratio becomes constant.

---

> ### Author Response · Authors · 2025-11-22
>
> ## Clarifications
>
> ### Figure 4 (Alignment Measurements)
> The alignment assumptions of µP are based on the mathematical formula for the gradients and properties of random weights (see Section 2.4, or e.g. Yang 2310.17813). These properties do not depend on the learning rate and thus whether µP's scaling is used or not. See also Section 5 / Appendix C for why the alignment can vary with width and time causing µP's alignment assumptions to fail.
>
> The specific measurements shown in Figure 4 do not use µP's scaling. However note the same behavior of the alignment ratio in Figure 3 which does include µP's scaling. We have also added Figure 23 in Appendix I that shows measurements of the alignment changes across multiple learning rates. Here we find that higher learning rates (around the optimal value) cause faster alignment changes than lower learning rates (which is still consistent with our hypothesis from Section 5).
>
> ### Warmup
> We agree that large early updates with linear warmup cause instabilities or lasting degradation that shifts the optimal learning rate to the left. This is fully consistent with our findings in this work, although we could not predict how quickly the alignment ratio would fall in practice. If the ratio were to fall sufficiently fast the linear warmup could have been sufficient to prevent large relative representation changes at any point. See our new Figure 19 for a case where the linear warmup does sufficiently prevent changes that are significantly larger than those seen later in training.
>
> By "surprisingly" we merely meant the empirical finding that a linear warmup is insufficient here despite being the default empirical approach used to train transformers and our setup here being fairly standard. The length of the warmup is often a consideration in practice, but as far as we know modifying the warmup shape is not common. We have modified this in the revision to make this clearer.
>
> ---
> ## Question Answers
>
> **Computation of $\|\Delta Y\|/\|Y\|$:** Yes this is computed at the same timestep / for the same input. We save the layer input and outputs on the forward pass, perform a standard backward pass and optimizer update, and then recompute the layer output based on the saved inputs.

---

### Official Review · Reviewer_wA1t · 2025-11-03

**Soundness:** 4
**Presentation:** 4
**Contribution:** 4
**Rating:** 6
**Confidence:** 4

**Summary:**

The paper studies the role of weight decay in learning rate transfer. In particular, the paper argues, both empirically and theoretically, that independent weight decay is crucial to learning rate transfer, i.e. the value of \eta * \lambda in PyTorch's AdamW implementation, should be fixed for different model widths. The key role of the independent weight decay is about controlling the relative update size of the feature. According the MuP's assumption at model initialization, the relative update size should scale with model widths, however the authors point out that as training proceeds, the assumption on independency between the  layer inputs and gradients, would break, and the relative update should become a constant value with respect to the model widths, which can be achieved by scaling the weight decay together with model width.

The authors then answers a couple of natural questions that arise, such as why would the assumption break and why we still need MuP scaling of learning rate anyway?

The paper conducts large scale empirical experiments on LLM pre-training under extensive hyperparameter sweep.

Overall I find the paper's claims and analysis convincing; the arguments and analysis angles from the paper can inform future works on hyperparameter transfer.

**Strengths:**

- The obesrvations and the arguments from the submission are not surprising, MuP's scalnig relies on lots of assumptions that don't hold and are unverified in later phases of the training, so it is not surprising that things will break. However, this paper is the first work, as far as I am concerned, that shows rigorously which ingredient in the MuP scaling breaks and why that is the case.

- The paper addresses most of the questions that naturally arise with the arguments, e.g. how and why the MuP assumptions break or why we still need MuP scaling of learning rate in practice.

- Although the paper involves lots of non-trivial technical details, I personally find the paper easy to follow, and the presentation of the results very clear.

- The experiments are fairly large-scale and convincing.

**Weaknesses:**

- It is unclear when the phase change will happen, and how does the phase changing will happen, does it have anything to do with the model size, the model width, etc.

- The paper does not seem to provide lots of new practical takeaways; the main conclusion seems to be that: One should use decoupled weight decay, which is already explicitly pointed out by many recent works.

- The paper seems only to argue that if this \eta * \lambda is **not fixed**, then learning rate transfer would break, i.e Eq.4 would start to depend on model widths. However, there is no direct argument on why learning rate becomes transferable as we fix \eta * \lambda.

- The paper is mostly around how to choose \eta, but does not show how to choose the value of \eta * \lambda or how to choose the value of \lambda.

**Questions:**

- Line 114, in the caption, should it be "... with a fixed absolute size |\Delta W|" rather than "|W|"?

- While I do undersand the argument intuitively, can the author clarify more on "... while a given relative change ∥∆W∥/∥W∥ always has the same impact ...", is "same impact" measured by the norm of \Delta Y, or is it measured by the value of Y?

- Could the fact that MuP with independent weight decay is needed because we want to restrict the absolute size of both the weights and the activations? I imagine under low precision LLM pre-training, we certainly want to weight norm and the activation norm. In particular, I believe the activation norm should depends on weight norm and the widths, and in order to prevent it from blowing up, we want to control the norm of W. Especially from the middle column in Fig.6, it seems that the consequence of not using MuP + independent weight decay is mostly on training stablity rather than the optimal learning rate starts to shift.

- If \eta * \lambda should be kept constant and \eta should scale with model width, do we have to scale \eta ~ 1 / m? Can we e.g. \eta as \eta_base / \sqrt(m) or \eta_base / m^p ?

- One question that I feel unanswered by the paper is: How should we pick the value of \eta * \lambda? The paper's experiments show that, if we keep \eta * \lambda fixed, then optimal \eta_base becomes transferable, but how should we choose the value of \lambda or the value of \eta * \lambda?

- It seems that there are two phase changes happening during the training, the first one is when the alignment ratio goes from width dependent to a value closer to 1, the second one is when the model entering the "equilibrium" and Eq.3 starts to hold, are these two phase changes happening at the same time?

- Why is the norm used for Frobenius norm rather than, e.g. Spectral norm?

---

> ### Author Response · Authors · 2025-11-22
>
> We thank Reviewer wA1t for their detailed review and positive assessment. We are encouraged that you found our analysis of the specific breakdown of µP ingredients rigorous and our experiments convincing. We have addressed your questions regarding the phase changes and practical takeaways in the updated manuscript (specifically adding Appendices G and H) and provide detailed responses below.
>
> ---
>
> ### W1: When and how do alignment changes happen?
> We hypothesize that the change in update alignment is driven by the evolution of the **Signal-to-Noise Ratio (SNR)** of the gradients. As detailed in **Section 5 and Appendix C**, our analytical model predicts that the update alignment depends on the interplay between the batch size, model width, and SNR.
>
> * **Early Training:** Gradients tend to be correlated across samples (high SNR) due to the network learning simple, frequent features (e.g., token frequency, common N-grams, simple grammar and syntax). This makes the overall batch gradient look more similar to the gradient of any given sample than when the gradients are uncorrelated. This is similar to having fewer but uncorrelated samples in the batch. The resulting update alignment looks similar to the small-batch regime assumed by µP.
> * **Late Training:** Once the model has learned these simple concepts, the remaining gradient for different samples involves more semantic nuance that varies significantly between samples. This results in a lower SNR and the overall update alignment becomes more similar to the uncorrelated large-batch regime, where we show the alignment becomes width-dependent $O(1/\sqrt{C})$.
>
> **We have added Figure 23** to empirically track these changes across different widths and learning rates.
>
> ### W2: Lack of New Practical Takeaways
> You are correct that our primary contribution is fundamental understanding, which significantly challenges prevailing beliefs about how learning rate transfer occurs in practice. We believe this understanding is valuable in identifying and resolving scaling issues, especially with novel optimizers or architectures. Beyond the recommendation for independent weight decay, our work offers two concrete directions:
>
> 1.  **Scaling for Fixed-Norm Training:** As an alternative to weight decay, we can instead constrain the weight norms to their initialization values using projections after each step. Although this approach is still not standard practice, it has shown promise for training diffusion models (Karras 2312.02696), reinforcement learning (Lyle 2407.01800) and transformer language model training (Loshchilov 2410.01131). In this case, there is no weight decay to counteract the learning rate scaling later in training. Our analysis suggests avoiding µP scaling here (or using explicit warmup instead). We discuss this in the new **Appendix H**. We believe similar approaches apply to relative optimizers like LARS (You 1708.03888), LAMB (You 1904.00962) and AdaFactor (Shazeer 1804.04235).
> 2.  **Exponential Warmup:** We show that exponential warmup schedules can replicate the benefits of µP, offering a tuning knob for stability that linear warmup lacks.
>
> ### W3: Why does the learning rate transfer when $\eta\lambda$ is fixed?
> Learning rate transfer works because scaling $\eta$ while fixing $\eta\lambda$ allows us to satisfy two conflicting requirements for the **relative weight update** ($\|\Delta W\|/\|W\|$) at different stages of training:
>
> 1.  **Early Training (Alignment $\approx$ Width-Dependent):** We need smaller initial updates for wider models to maintain stability under the high initial alignment ratios. Scaling $\eta \propto 1/m$ achieves this.
> 2.  **Late Training (Alignment $\approx$ Constant):** The alignment ratio becomes width-independent (approx. 1). To maintain constant relative representation changes, the relative weight update must also be constant across widths.
>
> Since the long-term relative update size is determined by the product $\eta\lambda$, fixing this product ensures the late-stage requirement is met, while scaling $\eta$ ensures the early-stage requirement is met.

---

> ### Author Response · Authors · 2025-11-22
>
> ### W4: How to choose the value of $\eta\lambda$?
> That is a fair limitation. Our work primarily addresses how the relative weight update $\|\Delta W\|/\|W\|$ should be scaled across width. We believe the primary effect of tuning $\eta$ for the base model is to find a good value for $\eta\lambda$, which in turn determines the size of the relative update throughout most of training. The weight decay value $\lambda$ only has to be somewhat reasonable for this to work.
>
> Regarding the specific choice of $\lambda$, while we cannot provide a single universal formula, we believe it should transfer to some extent when tuned for the base model. We have added **Appendix G** to explore how different pairs $(\eta, \lambda)$ with a fixed product affect training, including scheduling effects (similar to warmup) and how learnable gains can counteract changes in weight norms.
>
> Alternatively, one can fix the weight norms throughout training. This is a more principled approach that eliminates weight decay as a hyperparameter. In this setting, the learning rate directly controls $\|\Delta W\|/\|W\|$ and our main findings can be applied directly without confounding effects from weight decay. We have added **Appendix H** to explore this.
>
> ### Questions & Clarifications
> * **Q1 (Line 114):** Correct, this was a typo. We have fixed it to $\|\Delta W\|$.
> * **"Same Impact" of Relative Updates:** If the update direction is preserved, maintaining a constant relative update size results in the outputs being updated in exactly the same way across the rescaling transformations.
> * **Scaling $\eta \sim 1/m$ vs $1/\sqrt{m}$:** The $1/m$ scaling is necessary specifically because the initial alignment ratio scales with $\sqrt{m}$. If we used $1/\sqrt{m}$, the initial updates would be too large for wider models, likely causing instability. The $1/m$ scaling acts as the "correct" warmup to counteract this specific initial alignment. With sufficiently strong (likely exponential) warmup, one might be able to use $1/\sqrt{m}$ (similar to our Section 7 results).
> * **The effect of weight decay on activation norms:** Weight decay impacts weight norms, which in turn scale representation norms. However, architectures with learnable gains and normalization layers can counteract these changes. We explore this in **Appendix G** showing how the network learns to control query vector size using gains to counteract weight decay.
> * **Stability vs. Optimal LR Shift (Fig 6):** The optimal learning rate does not shift significantly in Figure 6 because the product $\eta\lambda$ is constant, maintaining the same relative update size throughout most of training. However, omitting µP scaling leaves early updates too large despite linear warmup (see Figures 16 and 19). These large updates degrade training, which explains the instability and the general need for learning rate warmup (Kosson 2410.23922).
> * **Low precision issues:** We do not believe numerical effects play a significant role in our stability results. We train in bfloat16, which has a dynamic range similar to float32. The difference in magnitude between representations and weights is relatively small (approx. 2 orders of magnitude; see Figure 17). For formats like fp8 or fixed point, these shifts could pose larger issues requiring explicit mitigation.
> * **Phase Changes:** The two phase changes happen simultaneously but are driven by different mechanisms. Alignment decay depends on the SNR and changes with meaningful learning, while equilibrium mechanics result from the geometry of the updates for a specific ($\eta, \lambda$) pair and occur even with random gradients.
> * **Spectral vs. Frobenius:** We focus on Frobenius norms because the theory of weight decay's effect on Frobenius norms is well-established, whereas the interaction with Spectral norms is less understood and harder to measure efficiently at scale.

---

> > ### Comment · Reviewer_wA1t · 2025-11-26
> >
> > Thanks for the response! My concerns are mostly addressed, but I have a couple of more minor comments, and have increased my score!
> >
> > ## learnable gain
> >
> > I went through the paper and didn't find the definition of learnable gain, but I believe this could correspond to the affine transformation parameter of normalization layer right? Maybe worth clarifying it somewhere.
> >
> >
> > ## The effect of weight decay on activation norms:
> >
> > It is worth pointing out that I don't think transformer is a fully scale invariant architecture (in contrast to ResNet, where all intermediate layers are scale invariant), there are lots layers in transformer that are not scale invariant, e.g. bottom left and bottom right grid in Figure 17, and my question is, does the magnitude of these activations matter if we want to study hyperparameter transfer?
> >
> >
> > -----------
> >
> > Overall, I think this is a great work!

---

### Meta-Review · Area_Chair_sUcC · 2026-01-06

**Summary:**

This paper studies learning-rate transfer under Maximal Update Parameterization (µP) and the role of independent weight decay. The paper introduces the concept of Relative Representational Change (RRC) to explain why µP’s alignment assumptions hold only early in training and why independent weight decay is necessary to maintain stable feature updates across different model widths. The authors provide a unified framework that combines theoretical analysis and empirical validation, showing that µP’s benefits largely arise from an implicit learning-rate warmup effect and that independent weight decay stabilizes training in later phases. Experiments span large-scale transformers (LLaMa) and ResNets, with comparisons to alternative warmup schedules that replicate µP’s effects.

**Reviewer Concerns:**

Some reviewers initially raised concerns about the clarity of the contribution, the choice of layers for alignment measurements, the justification of RRC over absolute representation changes, and the practical implications of the results. Additional questions addressed scaling choices, optimizer effects, and applicability to fine-tuning scenarios. The authors responded by adding explicit contribution bullets, additional figures and appendices showing RRC across multiple layers, performance metrics tables, and clarifications on methodology, warmup schedules, and weight decay effects. These revisions addressed all major concerns, demonstrating the robustness and reproducibility of the reported findings.

**Reviewer Scores:**

The submission received strong endorsements for its theoretical and empirical contributions. Key reviews rated soundness as excellent to fair (3–4), presentation as good to excellent (3–4), and contribution as fair to excellent (2–4). Overall, reviewers highlighted the novelty of introducing RRC, the practical relevance for hyperparameter transfer, and the clarity of experimental validation. After author responses clarifying methodology and adding supporting evidence, all major reviewers are now supportive of acceptance. Based on these assessments, I recommend acceptance.

---

### Decision · Program_Chairs · 2026-01-26

Accept (Poster)